# CX-5461 is a DNA G-quadruplex stabilizer with selective lethality in BRCA1/2 deficient tumours

Hong Xu[1], Marco Di Antonio[2,3], Steven McKinney[1], Veena Mathew[4], Brandon Ho[5], Nigel J. O'Neil[6], Nancy Dos Santos[7], Jennifer Silvester[8], Vivien Wei[1], Jessica Garcia[1], Farhia Kabeer[1], Daniel Lai[1], Priscilla Soriano[1], Judit Banáth[9], Derek S. Chiu[1], Damian Yap[1], Daniel D. Le[2], Frank B. Ye[6], Anni Zhang[4], Kelsie Thu[8], John Soong[10], Shu-chuan Lin[10], Angela Hsin Chin Tsai[1], Tomo Osako[1], Teresa Algara[1], Darren N. Saunders[1], Jason Wong[1], Jian Xian[11], Marcel B. Bally[7], James D. Brenton[11], Grant W. Brown[5], Sohrab P. Shah[1], David Cescon[8,12], Tak W. Mak[8], Carlos Caldas[11], Peter C. Stirling[4], Phil Hieter[6], Shankar Balasubramanian[2,3] & Samuel Aparicio[1]

G-quadruplex DNAs form four-stranded helical structures and are proposed to play key roles in different cellular processes. Targeting G-quadruplex DNAs for cancer treatment is a very promising prospect. Here, we show that CX-5461 is a G-quadruplex stabilizer, with specific toxicity against BRCA deficiencies in cancer cells and polyclonal patient-derived xenograft models, including tumours resistant to PARP inhibition. Exposure to CX-5461, and its related drug CX-3543, blocks replication forks and induces ssDNA gaps or breaks. The BRCA and NHEJ pathways are required for the repair of CX-5461 and CX-3543-induced DNA damage and failure to do so leads to lethality. These data strengthen the concept of G4 targeting as a therapeutic approach, specifically for targeting HR and NHEJ deficient cancers and other tumours deficient for DNA damage repair. CX-5461 is now in advanced phase I clinical trial for patients with BRCA1/2 deficient tumours (Canadian trial, NCT02719977, opened May 2016).

[1] Department of Molecular Oncology, British Columbia Cancer Research Centre, and Department of Pathology and Laboratory Medicine, University of British Columbia, 675 West 10th Avenue, Vancouver, British Columbia, Canada V5Z 1L3. [2] Cancer Research UK Cambridge Research Institute, Li Ka Shing Centre, Robinson Way, Cambridge CB2 0RE, UK. [3] Department of Chemistry, University of Cambridge, Cambridge CB2 1EW, UK. [4] Terry Fox Laboratory, BC Cancer Agency, 675 West 10th Avenue, Vancouver, British Columbia, Canada V5Z 1L3. [5] Department of Biochemistry and Donnelly Centre, University of Toronto, 160 College Street, Toronto, Ontario, Canada M5S 3E1. [6] Michael Smith Laboratories, University of British Columbia, Vancouver, Canada V6T 1Z4. [7] Advanced Therapeutics, BC Cancer Agency and Department of Pathology and Laboratory Medicine, University of British Columbia, 675 West 10th Avenue, Vancouver, British Columbia, Canada V5Z 1L3. [8] Campbell Family Institute for Breast Cancer Research, Princess Margret Cancer Centre, 610 University Avenue, Toronto, Canada M5G 2M9. [9] Department of Integrative Oncology, BC Cancer Agency, 675 West 10th Avenue, Vancouver, British Columbia, Canada V5Z 1L3. [10] Senhwa Biosciences, Inc., 9 F, No.205-1, Section 3, Peihsin Road, Hsintien District, New Taipei City 23143, Taiwan R.O.C. [11] Cancer Research UK Cambridge Research Institute and Department of Oncology, University of Cambridge, Li Ka Shing Centre, Cambridge CB2 0RE, UK. [12] Division of Medical Oncology and Hematology, Department of Medicine, University of Toronto, Toronto, Canada M5S 1A8. Correspondence and requests for materials should be addressed to S.A. (email: saparicio@bccrc.ca).

Inherited *BRCA2* mutations predispose carriers to early onset breast, ovarian and other cancers[1,2]. As an important tumour suppressor, the key role of BRCA2 is in homologous recombination (HR)-mediated DNA damage repair by promoting the formation of RAD51 filaments at DNA breaks[3]. When BRCA2 is deficient, HR repair efficiency is greatly compromised, leading to an increased error-prone DNA repair and ultimately, genomic instability. The BRCA2/RAD51 complex is also involved in many other aspects of genome instability, such as stalled DNA replication fork stabilization[4], R-loop resolution and repairing G-quadruplex (G4) associated DNA damage[5,6].

G4 structures can potentially form at over 700,000 sequences in the human genome[7,8], and 10,000 of them have been identified from ChIP-seq using an antibody that recognises G4 structures[9]. G4 structures increase the tendency for DNA damage to occur, by impeding DNA polymerase and DNA damage repair processes[10]. The importance of the HR pathway in repairing G4-induced DNA damage has been demonstrated in different organisms[11,12]. BRCA2-deficient cells display higher sensitivity to tool compounds such as pyridostatin (PDS)[13] and RHS4 (ref. 6), which are both G4 stabilizers, but not medicinal compounds and are structurally unrelated to the fluoroquinolone derived CX series compounds.

As a general phenomenon in cancer, there is an increased requirement for rDNA transcription to meet the greater protein synthesis demand in cancer cells[14]. Some new inhibitors of rDNA transcription have been synthesized in recent years, such as CX-5461, CX-3543 and BMH-21 (refs 15–17). CX-3543 binds to G4 sequences and disrupts the interaction of rDNA G4 structures with nucleolin, thereby inhibiting Pol I transcription and inducing apoptotic death in cancer cells[16]. BMH-21 acts by its interaction with the DNA backbone in GC-rich DNA sequences, particularly at rDNA loci, thus inhibiting Pol I transcription and also promoting degradation of Pol I catalytic subunit RPA194 (ref. 18). CX-5461 is an rDNA transcription inhibitor currently in phase I trials for haematologic malignancies. CX-5461 reduces the binding affinity of the SL1 pre-initiation complex and RNA polymerase I complex to rDNA promoters and conveys p53-dependent anti-tumorigenic activity in hematopoietic malignancies[15,17]. Recently, more targets of CX-5461 have been discovered, such as the activation of ATM/ATR[19] and rapamycin-associated signalling pathway[20].

In the present study, we have uncovered a new and unanticipated mechanism of CX-5461 activity in HR and non-homologous end joining (NHEJ) deficient cancer cells. We show that both CX-5461 and the related compound CX-3543 induce DNA damage and are dependent on BRCA1/2-mediated HR and DNA-PK-mediated NHEJ pathway for damage repair. We also discover that CX-5461 (and CX-3543) bind and stabilize G4 DNA structures *in vitro*, impede the progression of DNA replication complexes and result in increased *in vivo* G4 structures. The pattern of activity in polyclonal patient-derived xenografts (PDX) mirrors that seen *in vitro* with isogenic cell line pairs, namely sensitivity in BRCA deficient PDX models, in the context of pre-treatment with taxane and other standard of care agents. In some cases, superior activity to PARP inhibition is observed. Our data suggest that the CX drugs, and possibly other G4 stabilizers have the potential to treat cancers deficient for BRCA1, BRCA2, NHEJ pathway members and some other genes involved in DNA damage repair and DNA replication. Since CX5461 is an advanced phase I medicinal compound, these observations have immediate translational significance.

## Results
### CX-5461 selectively inhibits cancer cells deficient for BRCA1/2.
To identify potential novel drugs for cancers with *BRCA2*

mutations, we tested a total of 17 commercially available inhibitors (Supplementary Table 1) by clonogenic assays in isogenic BRCA2 knockout and wild type (WT) HCT116 cell line pairs published by us[21]. This clonogenic screen identified CX-5461, a previously described RNA pol I inhibitor[15,17] to be highly toxic to BRCA2 knockout HCT116 cells as compared with isogenic BRCA2 WT cells (Fig. 1a). We extended the quantification of this observation by using a WST-1 metabolic/cell viability assay. As with the clonogenic assay, this revealed a 9.0-fold (95% confidence interval (CI), 5.1–16.2) lower $IC_{50}$ in BRCA2 deficient HCT116 cells than in BRCA2 proficient cells (Fig. 1b, Supplementary Fig. 1a). Importantly, we observed in this experiment and those described below, that *BRCA2* heterozygous cells displayed similar sensitivity to CX-5461 as BRCA2 proficient wild-type cells (Fig. 1b,d). We also assessed cell death specifically through fluorescence-activated cell sorting (FACS) by annexin V and PI double staining. As shown in Fig. 1c and Supplementary Table 5, CX-5461 induced more apoptotic cell death in *BRCA2* knockout cells relative to WT. However, $BRCA2^{+/+}$ and $BRCA2^{-/-}$ isogenic cells in HCT116 appeared equally sensitive to actinomycin (an inhibitor for both RNA polymerase I and II) and cycloheximide (an inhibitor for protein translation elongation) (Supplementary Fig. 1b,c). Together, these data indicate that BRCA2 deficient cells are not generally sensitive to transcription and translation inhibition, but show specific sensitivity to CX-5461.

We next sought to determine whether the selective killing effect of CX-5461 in BRCA2 deficient cells could be observed in other cell lines and species backgrounds. We measured CX5461 drug sensitivity in isogenic $BRCA2^{-/-}$ and WT colorectal cancer DLD1 cells; BRCA2 deficient ovarian cancer cell line PEO1 *vs* the BRCA2 proficient C4-2 (ref. 22); and BRCA2 proficient and knockout breast tumour cells derived from p53 knockout mouse ($BRCA2^{-/-}$; $p53^{-/-}$ (KB2P 1.21) *vs* $BRCA2^{+/+}$; $p53^{-/-}$ (KP6.3) cells)[23]. In all these cell line pairs, BRCA2-deficient cells were more sensitive to CX-5461 than BRCA2 proficient cells (Fig. 1d,e, Supplementary Figs 1d and 2a–c). When BRCA2 was transiently knocked down in U2OS, increased sensitivity to CX-5416 was also observed (Supplementary Fig. 2d).

In $p53^{-/-}$ HCT116 cells, drug sensitivity to CX-5461 increased significantly (*F*-test $P = 0.01$) after BRCA2 knockdown (Fig. 1f). This supports the notion that the selective toxicity of CX-5461 to BRCA2 deficient cells is not dependent on p53 in epithelial cells[17]. BRCA1 is another critical component of the HR repair pathway and is also involved in replication fork stabilization[24,25]. Encouragingly, increased drug sensitivity to CX-5461 was observed when BRCA1 was knocked down in $p53^{+/+}$ and $p53^{-/-}$ HCT116 (Supplementary Fig. 2e), further supporting that synthetic lethality with CX-5461 does not require wild type p53. We also note (below) that CX-5461 sensitivity is seen in polyclonal p53 deficient PDX tumours (model CFIB-70620).

### BRCA2 is not synthetic lethal with rDNA transcription inhibition.
We set out to address the possible molecular mechanism by which CX-5461 causes synthetic lethality in BRCA2 deficient cells. Since CX-5461 has documented RNA pol I inhibition activity, we investigated this notion first, by examining the effects of a related compound CX-3543 (quarfloxin) and an unrelated small molecule BMH-21. Both of these molecules are reported to suppress rDNA transcription, but by different mechanisms[16,18]. In DLD1 cells, $BRCA2^{-/-}$ cells were highly sensitive to CX-3543 compared with isogenic $BRCA2^{+/+}$ cells (Supplementary Fig. 3d). In contrast, $BRCA2^{-/-}$ cells were not more sensitive to BMH-21 than isogenic WT cells in either HCT116 or DLD1 (Fig. 1h, Supplementary Fig. 3a,b).

Since BRCA2 deficient cells showed differential response to the three rRNA transcription inhibitors, we sought to verify the activity against rDNA transcription in the cells used. We measured the amount of 45S pre-rRNA as a readout for rDNA transcription by qRT-PCR, as previously described[16], at 2 and 24 h after treatment. In these experiments, BMH-21 was at least as potent an rDNA transcription inhibitor as CX-5461 and CX-3543 (Fig. 1g, Supplementary Fig. 3g), consistent with published activity[18]. However, the same or greater level (median 1–2 fold for CX-5461, CX-3543 at $10^{-6}$ M $vs$ median

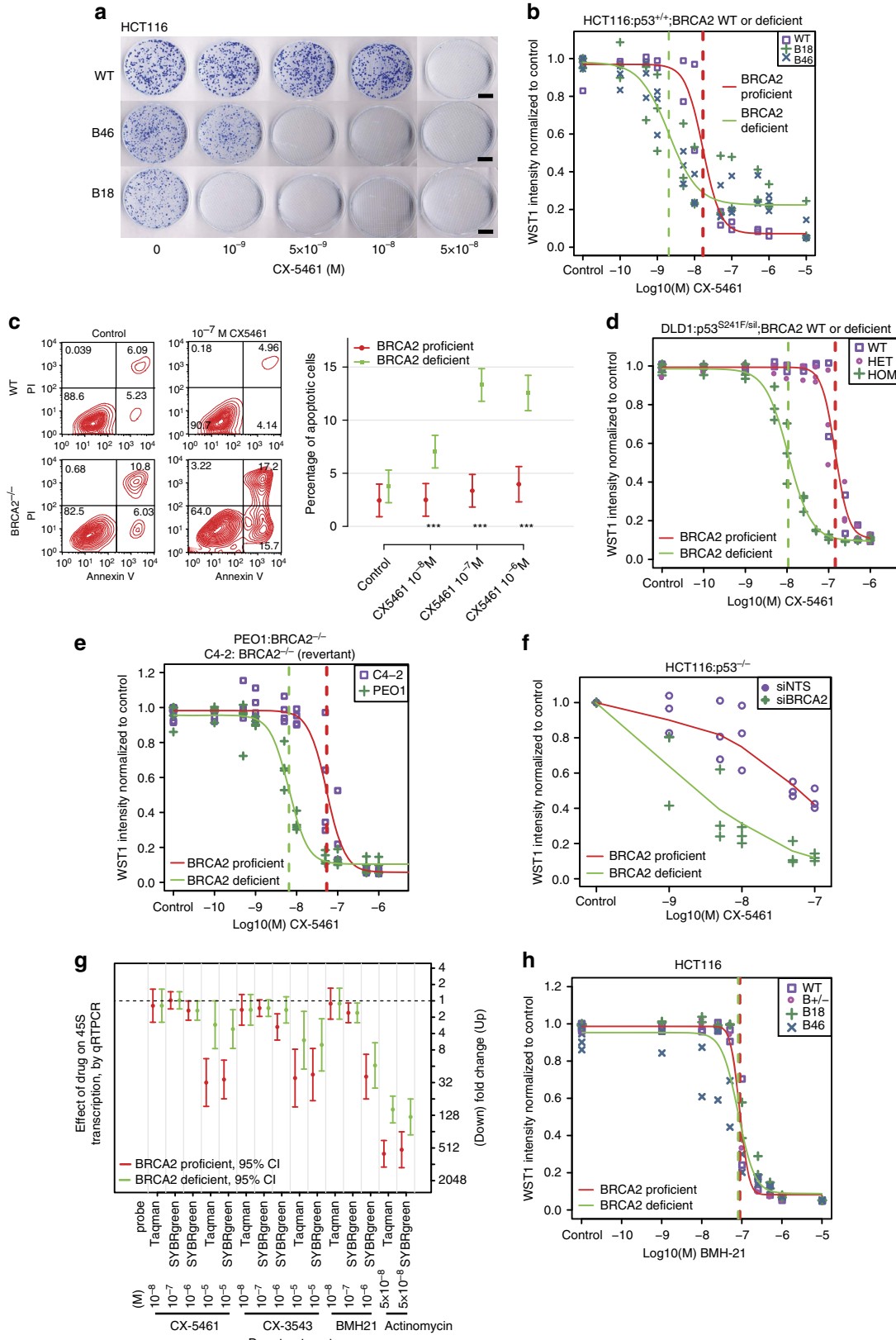

16-fold for BMH-21 at $10^{-6}$ M, Fig. 1g, Supplementary Fig. 3g) of rDNA transcription inhibition by BMH-21 did not result in selective toxicity against $BRCA2^{-/-}$ cells, in contrast with CX-5461 and CX-3543. We also noted that the IC$_{50}$ of CX-5461 in the cell lines used is $4.80 \times 10^{-9}$ M (Fig. 1b, Supplementary Fig. 1a) for BRCA2 knockout HCT116 cells, a concentration at which rDNA transcription inhibition was not observed. We thus concluded that rDNA transcription inhibition alone is not required for the selective toxicity of CX-5461 and CX-3543 against BRCA2 deficient cells.

We also compromised rDNA transcription with RNAi against POLR1B, an essential component in the RNA pol I complex. Upon POLR1B knockdown, rDNA transcription was significantly reduced (Supplementary Fig. 3c), and cell viability also greatly decreased by single RNA pol I knockdown, making measurement of synergy with BRCA2 (double knockdown) difficult. In U2OS cells, POLR1B and BRCA2 double knockdown reduced cell viability at a level comparable with single POLR1B knockdown (Supplementary Fig. 3f), with non-statistical difference seen, consistent with the absence of a synthetic genetic effect in U2OS. In HCT116, slightly more cell death was observed in double knockdown cells compared with single knockdown cells (omnibus test $P < 0.0001$) (Supplementary Fig. 3e), however the single RNA pol I knockdown induces many fold greater cell death than the knockdown of BRCA2, precluding accurate assessment of synthetic lethality. Taken together with the results of BMH-21, the data disfavour rDNA transcription inhibition as an important mechanism of CX-5461 toxicity in BRCA2 deficient cells.

When drug sensitivity of CX-5461, CX-3543, and BMH-21 was compared across 50 breast cancer cell lines, the response signature of BMH-21 was different from CX-5461 and CX3543 (Fig. 7d, Supplementary Table 4). BMH-21 is only moderately cytotoxic to a few breast cancer cell lines, while CX-3543 has response pattern very similar to CX-5461 and is clustered together with CX-5461 (Fig. 7d), consistent with a similar mode of action between CX-3543 and CX-5461, but distinct in the case of BMH-21.

**CX-5461 and CX-3543 induce DNA damage in tumour cells.** As the major cellular functions of BRCA2 are DNA damage repair and replication fork stabilization, we investigated DNA damage response upon exposure to CX-5461, CX-3543 and BMH-21. Strikingly, $\gamma$-H2AX and 53BP1 DNA damage foci were robustly induced by CX-5461 and CX-3543 in both HCT116 and U2OS cells when the drug concentration was higher than $10^{-8}$ M (Fig. 2a–c). In marked contrast, no $\gamma$-H2AX or 53BP1 focus formation was visible with $10^{-8}$ M and $10^{-7}$ M BMH-21,

although $10^{-6}$ M BMH-21 induced a statistically significant level of damage foci that was slightly higher (Fig. 2b,c). Consistent with the result of DNA damage protein focus formation, increased phosphorylation of H2AX was observed after CX-5461 and CX-3543 treatment by Western blotting analysis (Fig. 4a, Supplementary Fig. 4a). As shown above, BMH-21 inhibited rDNA transcription more potently than CX-5461 and CX-3543 (Fig. 1g). However, BMH-21 did not appear to be associated with the same degree of DNA damage and cell death as CX-5461 and CX-3543 at the same concentration. Moreover, reduction of rDNA transcription by siRNA for POLR1B did not result in more $\gamma$-H2AX and 53BP1 foci formation (Supplementary Fig. 4b). Collectively, these data strongly suggest that CX-5461 and CX-3543 are potent DNA damage inducers, and this mechanism is independent of rDNA transcription inhibition in the cell types examined. We also excluded the possibility that the DNA damage phenotype was an indirect effect of nucleolus disruption, because DNA damage foci were observed when the concentration of CX-5461 was not able to disrupt nucleolus (Supplementary Fig. 4e).

In order to better quantify the amount and type of DNA damage, a comet tail forming assay[26] was applied to short exposure CX-5461 treated cells. After 30 min of drug exposure, statistically increased comet tail-moments were observed under alkaline conditions when the concentration of CX-5461 was higher than $10^{-7}$ M ($P < 10^{-15}$, $\chi^2$ test for all drug concentrations tested) (Fig. 2d,e). Under neutral comet assay conditions, which are more specific for DSBs, we discovered a small but statistically significant increase in tail moments ($P < 10^{-6}$, $\chi^2$ test for all drug concentrations, Fig. 2d). Thus, upon short exposure (30 min) to CX-5461, the initial forms of DNA damage occurring are mainly SSBs or gaps and a lower abundance of DSBs.

Remarkably, CX-5461 (Supplementary Fig. 4d) also induced DNA damage foci in yeast and selectively killed yeast lacking RAD52 (Supplementary Fig. 4h), a functional homologue of BRCA2, further supporting DNA as the drug target of CX-5461 and CX-3543.

**CX-5461 and CX-3543 induced DNA damage is replication-dependent.** To further investigate the mechanism of CX-5461 and CX-3543 induced DNA damage, cell cycle analysis was performed on drug treated HCT116 cells by FACS with EdU and PI staining. Shortly after $10^{-6}$ M CX-5461 treatment (2 h), active replication shown by EdU staining decreased significantly (7.1% decrease; 95% CI, 2.6–11.5%) in $BRCA2^{+/+}$ cells (Fig. 3a). At a later time point (24 h after), CX-5461 induced a prominent

**Figure 1 | BRCA2 deficient cells are highly sensitive to CX-5461 in different human and murine cell types.** (**a**) The colony formation capacity of $BRCA2^{-/-}$ HCT116 cells was greatly reduced by treatment with CX-5461. Experiments were repeated twice with similar results. Scale bar, 1 cm. (**b**) The hypersensitivity of $BRCA2^{-/-}$ cells to CX-5461 in HCT116 validated by WST-1 assay. Representative experiment #3 (see Supplementary Fig. 1a for full experimental panels, $n = 9$) is displayed as individual data points and as fitted sigmoid dose response curves (green for BRCA2 deficient and red for BRCA2 wild type). Dashed vertical lines are the IC50. (**c**) CX-5461 induced more apoptosis in $BRCA2^{-/-}$ cells as indicated by FACS analysis. A representative result is shown on the left panel and the right panel shows mean apoptotic fraction with 95% CI of cells in early apoptosis under different drug concentrations. ***$P < 0.0001$, $t$-test; $n = 3$, 2 or more replicates per condition. See Supplementary Table 5 for more statistical analyses. (**d**) $BRCA2^{-/-}$ in DLD1 isogenic cell line pairs displayed hypersensitivity to CX-5461 by WST-1 assay. Representative experiment #3 is shown (see Supplementary Fig. 2a for full experimental panels; $n = 4$). Green fitted sigmoid curves are for $BRCA2$ homozygous (HOM) and red for $BRCA2$ wild type and heterozygous (HET). (**e**) BRCA2 deficient ovarian cancer PEO1 cells exhibited increased sensitivity to CX-5461 relative to BRCA2 proficient C4-2 cells by WST-1 assay. A representative experiment #1 is displayed (see Supplementary Fig. 2b for full experimental panels, $n = 3$). (**f**) RNAi knockdown of BRCA2 increased sensitivity to CX-5461 in $p53^{-/-}$ HCT116 cells by WST-1 assay (4 days in drug). The results of all three experiments are summarized by the green (BRCA2 knockdown) and red (non-targeting control) super-smoother fit lines. (**g**) 45S pre-rRNA level measured by RT-PCR after CX-5461, CX-3543 and BMH-21 treatment in WT and $BRCA2^{-/-}$ HCT116 cells. Drug incubation time was 24 h. Fold change estimates and unadjusted 95% CIs of 45s pre-rRNA levels under drug treatment condition versus vehicle control are shown. $P$ values (by $F$-test) are shown in Supplementary Table 7. (**h**) BRCA2 knockout cells are not more sensitive to BMH-21 in HCT116 through WST-1 assay. One representative experimental result is shown (more replicates are shown in Supplementary Fig. 3a).

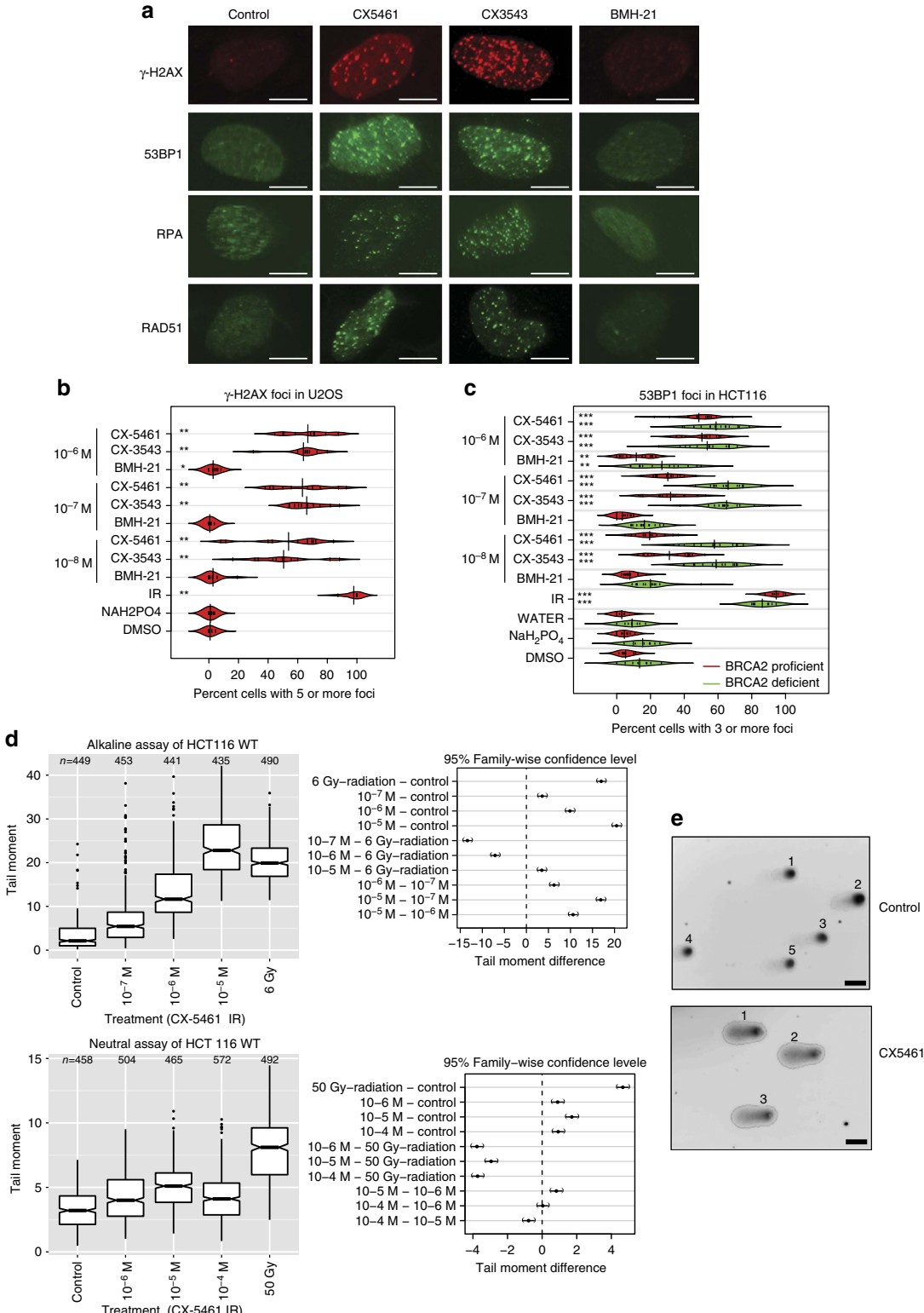

**Figure 2 | DNA damage is induced in cells with CX-5461 and CX-3543 treatment.** (**a**) The formation of γ-H2AX, 53BP1, RPA and RAD51 foci was monitored upon CX-5461, CX-3543 and BMH-21 treatment at $10^{-7}$ M in U2OS cells. Drug treatment time is 24 h for all drugs. Scale bar, 10 μM. (**b**) Beanplots of U2OS cells showing five or more γ-H2AX foci for the indicated drug treatment condition after 24 h. **$P < 0.01$ (one tailed randomization tests adjusted for multiple comparisons relative to vehicle control); $n = 2$ experiments, (35–150 cells were counted each time per each condition). DMSO was the vehicle control solvent for CX-3543 and NaH$_2$PO$_4$ was the vehicle control solvent for BMH-21 and CX-5461. (**c**) Beanplots of HCT116 cells showing three or more 53BP1 foci for the indicated drug treatment conditions after 24 h. **$P < 0.01$, ***$P < 0.0001$ (one-tailed randomization tests adjusted for multiple comparisons relative to vehicle control); $n = 3$ experiments; >100 cells per replicate condition. (**d**) Quantification of alkaline comet and neutral comet assay result upon CX-5461 treatment (30 min) in HCT116. Tail moments were determined as described in Methods; ($n = 3$ experiments, 100 cells were counted in each replica), mean and 95% CI are shown. Right panels show pair-wise comparison of condition contrasts and 95% CIs of tail moment difference. (**e**) Representative images of alkaline comet assay of cells treated with CX-5461 ($10^{-6}$ M, 30 min) and no drug control. Scale bar, 20 μM.

G2 arrest (41% increase; 95% CI, 37–45%), blocking cells before mitosis (Supplementary Fig. 5b). In HCT116 $BRCA2^{-/-}$ cells, the decrease of S-phase population was more dramatic (10.2% decrease; 95% CI, 5.6–14.9%, 2 vs 0 h, Fig. 3a, Supplementary Fig. 5a, Supplementary Table 6).

To determine whether the DNA damage was occurring in S-phase, we blocked DNA replication with aphidicolin (APH), a DNA polymerase inhibitor. Strikingly, for EdU positive S phase cells, APH pre-treatment remarkably reduced the percentage of cells with DNA damage foci relative to CX-5461 only treatment in both WT (decreased 48.6%; 95% CI, 44.2–51.2%) and $BRCA2^{-/-}$ cells (decreased 68.5%; 95% CI, 57.1–75.2%) (Fig. 3b,c). But for EdU negative cells, APH treatment showed no effect. The predominance of DNA damage foci in S-phase population and its dependence on DNA replication were also observed with CX-3543 treatment (Supplementary Fig. 5c,d). Likely, CX-5461 and CX-3543 bind to DNA and impede DNA replication forks, with the consequence that non-resolved stalled forks give rise early on to ssDNA gaps and breaks. In line with this model, RPA complex, which specifically accumulates at ssDNA tracts, formed foci after CX-5461 and CX-3543 treatment (Fig. 2a, Supplementary Fig. 4f), and the ATR–CHK1 pathway was activated as revealed by increased phosphorylation of CHK1 and CHK2 with CX-5461 treatment (Supplementary Fig. 4c). Collectively, these data suggest that CX-5461 and CX-3543 induce replication-dependent DNA damage.

To better define the nature of this replication defect caused by CX-5461/CX-3543, DNA fibre analysis was utilized to monitor DNA replication at a single molecule level. In this procedure, nascent replication tracks were labelled before (green CIdU labelling) and after (red IdU labelling) CX-5461 treatment to evaluate its immediate effect (30 min) on replication. CX-5461 substantially shortened replication track length in both $BRCA2^{+/+}$ and $BRCA2^{-/-}$ HCT116 cells, and this effect was more prominent in $BRCA2^{-/-}$ cells (Fig. 3d). Significant fork rate reduction was observed in the presence of CX-5461 at a concentration 10 times lower in $BRCA2^{-/-}$ cells than in $BRCA2^{+/+}$ cells. Thus, CX-5461 blocks replication fork progression and BRCA2 functions in bypassing the replication obstacles generated by CX-5461.

**CX-5461 and CX-3543 induced DNA damage is repaired by HR pathway.** In the absence of functional BRCA2, a higher percentage of cells exhibited 53BP foci upon CX-5461 and CX-3543 treatment than in BRCA2 proficient cells (Fig. 2c). Furthermore, stronger γ-H2AX and RPA phosphorylation were observed in $BRCA2^{-/-}$ cells compared to $BRCA2^{+/+}$ cells by Western blotting (Fig. 4a). RPA phosphorylation happened at RPA2-pT21, a site associated with replication stress[27]. These findings suggest that DNA damage is less efficiently repaired in BRCA2 deficient cells. Moreover, RAD51 foci were detected upon CX-5461 and CX-3543 treatment (Fig. 2a). Most RAD51 foci co-localized with γ-H2AX foci, implying that the BRCA2/RAD51 complex localizes to damaged DNA and is directly involved in repairing DNA damage generated by CX-5461 and CX-3543 (Supplementary Fig. 5e).

To evaluate whether CX5461-induced cell death in BRCA2 knockout cells is caused by unrepaired DNA damage, mitotic chromosome spreads were examined to visualize chromosome breaks and chromosome structure abnormalities. A 48 h treatment with $10^{-8}$ M CX-5461 had no effect on WT cells, but significantly induced more chromosomal abnormalities in $BRCA2^{-/-}$ HCT116 (Fig. 4b,c). Likely, the unrepaired DNA damage kills $BRCA2^{-/-}$ cells. We further compared the kinetics of repairing CX-5461 associated DNA damage in $BRCA2^{+/+}$

and $BRCA2^{-/-}$ cells. We pulse treated cells with $10^{-8}$ M CX-5461 for 2 h, then washed out drug and examined DNA damage recovery. In WT, 72 h after drug pulse treatment, the percentage of cells with 53BP1 foci was significantly reduced in comparison with their initial induction at 2 h (Fig. 4d,e). In BRCA2 knockout cells, 53BP1 foci were not significantly reduced at 72 h and remained higher than no-drug baseline. These results suggest the important role of BRCA2 in repairing CX-5461 induced DNA damage, and that compromised DNA damage repair in the absence of BRCA2 will lead to lethality.

**CX-5461 is a G-quadruplex stabilizer in the human genome.** CX-3543 has been shown to have G4 binding/stabilizing activity[16]. This prompted us to examine whether CX-5461, which has a related structure, is capable of binding and stabilizing G4 forming sequences. We performed a FRET-melting assay with these compounds using DNA oligonucleotides comprising three different G4 forming sequences (c-MYC, ckit1 and h-Telo)[28] and a control dsDNA sequence. In these experiments a known G4 binding and stabilizing small molecule, PDS[29], was used as a positive control. Both CX-5461 and CX-3543 displayed an increased melting temperature ($\Delta T_m$) ($>15\,°C$) in the presence of 1 μM of either compound (Fig. 5a). Conversely, poor stabilization and non-specific binding profiles were recorded when treating a dsDNA forming sequence with CX-5461 or CX-3543 (Fig. 5a), suggesting that both compounds can selectively bind and stabilize G4 structures over the canonical double helical DNA. In addition, the stabilization of the G4 structure by these compounds was not affected in the presence of competitor dsDNA up to 50 mol equiv. of excess (Supplementary Fig. 6a,b), further confirming that CX-5461 and CX-3543 are selective G4 binders. In contrast, BMH-21 revealed no detectable G4 binding or stabilization (Fig. 5a). To assess the ability of stabilized G4 sequences to stall a DNA polymerase, we performed an in vitro DNA polymerase extension/processivity assay[30]. Strikingly, same as PDS, both CX-5461 and CX-3543 increased DNA polymerase stalling selectively at the G4 site in the cKit1 template (Fig. 5b).

Next, we investigated whether CX-5461 and CX-3543 have G4 stabilization properties in a cellular environment, by performing immunofluorescence with the G4 selective antibody BG4 (ref. 31) in HCT116 WT cells after incubation with 100 nM CX-5461 or CX-3543 for 24 h. Notably both CX-3543 and CX-5461 showed a significant increase of nuclear BG4 foci (Fig. 5c), suggesting that both compounds can trap and stabilize G4 structures in vivo at nanomolar concentrations. We also measured the co-localization of DNA damage 53BP1 foci and BG4 foci with and without CX-5461/CX-3543, and found significantly increased co-localization in the presence of CX drugs and PDS in contrast to no drug control and doxorubicin treatment (Fig. 5d).

To test directly whether chromosome destabilization by CX-5461 is dependent on G4 structures, we performed a modified gross chromosomal rearrangement (GCR) assay in yeast[32], with a known G4 DNA prone site, or a non-G4 forming G-rich control sequence inserted near the selectable markers (Fig. 6a). By using a sensitized background bearing the pif1-m2 allele, we found CX-5461 significantly increased GCR events compared to the G-rich but non-quadruplex-forming control (Fig. 6a). Untreated cells were not significantly different from each other. In a human cell system, we investigated the effect of CX-5461 on the integrity of telomeres, loci enriched with G4 structures. Telomere FISH results show an increased frequency of telomere defects in both $BRCA2^{+/+}$ and $BRCA2^{-/-}$ HCT116 cells after exposure to CX-5461, and this defect was more prominent in $BRCA2^{-/-}$ cells (Fig. 6b). Collectively, these data support CX-5461 as a G4

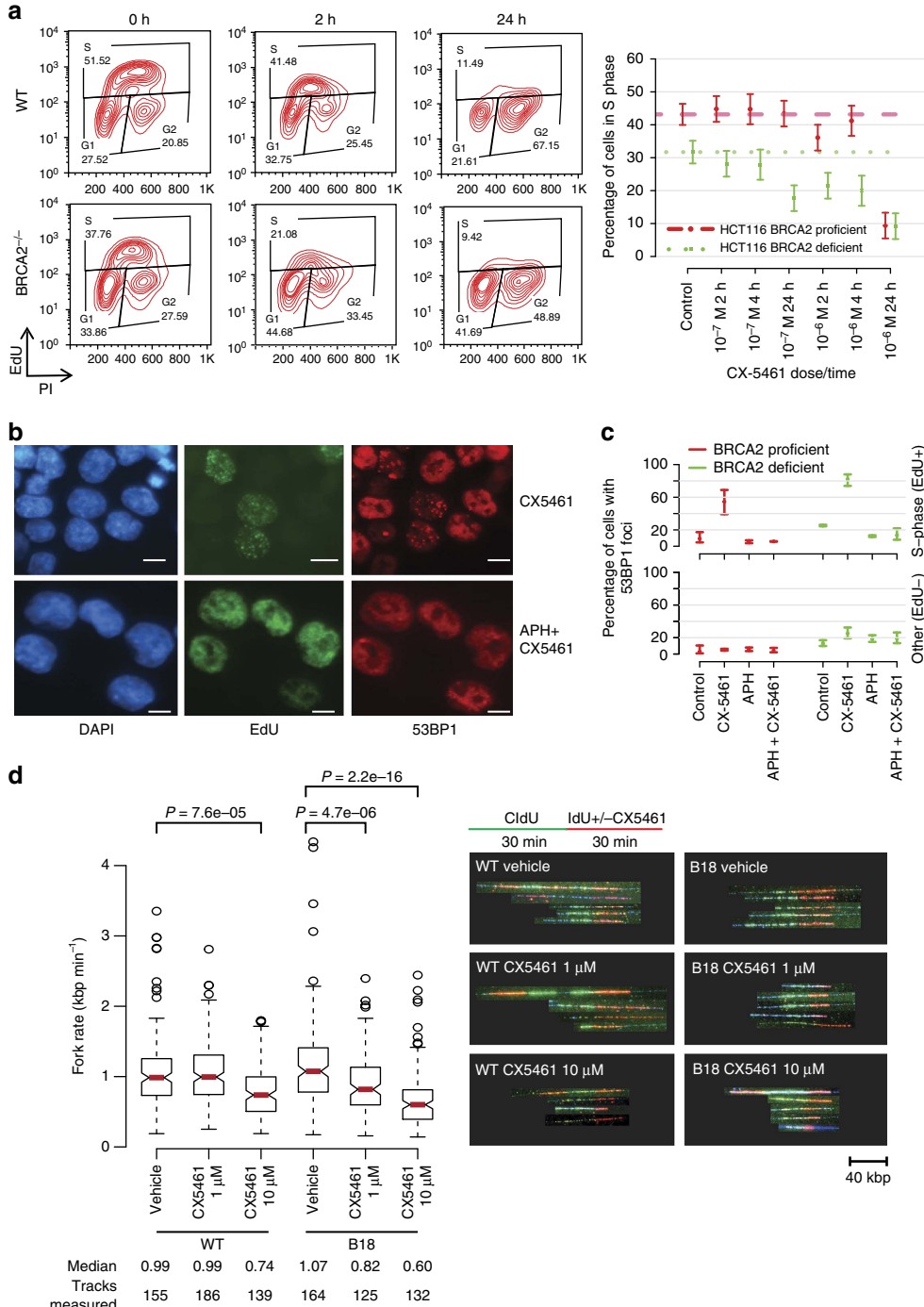

**Figure 3 | CX-5461 and CX-3543 induced DNA damage is replication-dependent. (a)** Active replication decreased upon CX-5461 treatment in WT and *BRCA2*[−/−] HCT116. Cells were treated with CX-5461 for the time indicated before incubating with EdU (10 μM) for 1 h. Cells were analysed by FACS with the intensity of EdU and PI recorded. Left panel shows one representative FACS profile when cells were treated with CX-5461 at 10[−6] M; right panel shows the mean percentage of cells in S phase (with 95% CIs) under different CX-5461 concentrations at different time points; $n = 3$ experiments. Cell cycle distributions at more time points and drug concentrations are shown in Supplementary Fig. 5a and Supplementary Table 6. **(b)** CX-5461 induced 53BP1 foci enriched in S phase (positive for EdU labelling), and APH greatly suppressed CX-5461 induced DNA damage in HCT116. WT Cells were treated with EdU (20 μM) for 30 min, then EdU was washed out and the cells were treated with CX-5461 (10[−7] M) for 1 h. For APH treatment, after EdU labelling, APH (5 μM) was added for 2 h before incubating with CX-5461 (10[−7] M) for 1 h. Scale bar, 10 μM. **(c)** The percentage of 53BP1 foci positive cells within EdU positive and EdU negative population with or without APH was quantified in HCT116 cells. Experimental conditions were the same as stated in **b**. Bars show the mean of three time course experiments (>100 cells each replica) and 95% CIs. **(d)** Replication rate is reduced by CX-5461 in BRCA2 deficient cells at higher level than in BRCA2 proficient cells. CldU (30 min) treated HCT116 cells were chased with or without CX-5461 for 30 min in the presence of IdU, then the cells were processed for DNA fibre analysis; $n = 2$. Median fork rate and the number of tracks analysed are shown. The box extends from the 25th to 75th percentiles. *P* value was calculated by Mann–Whitney *U* test.

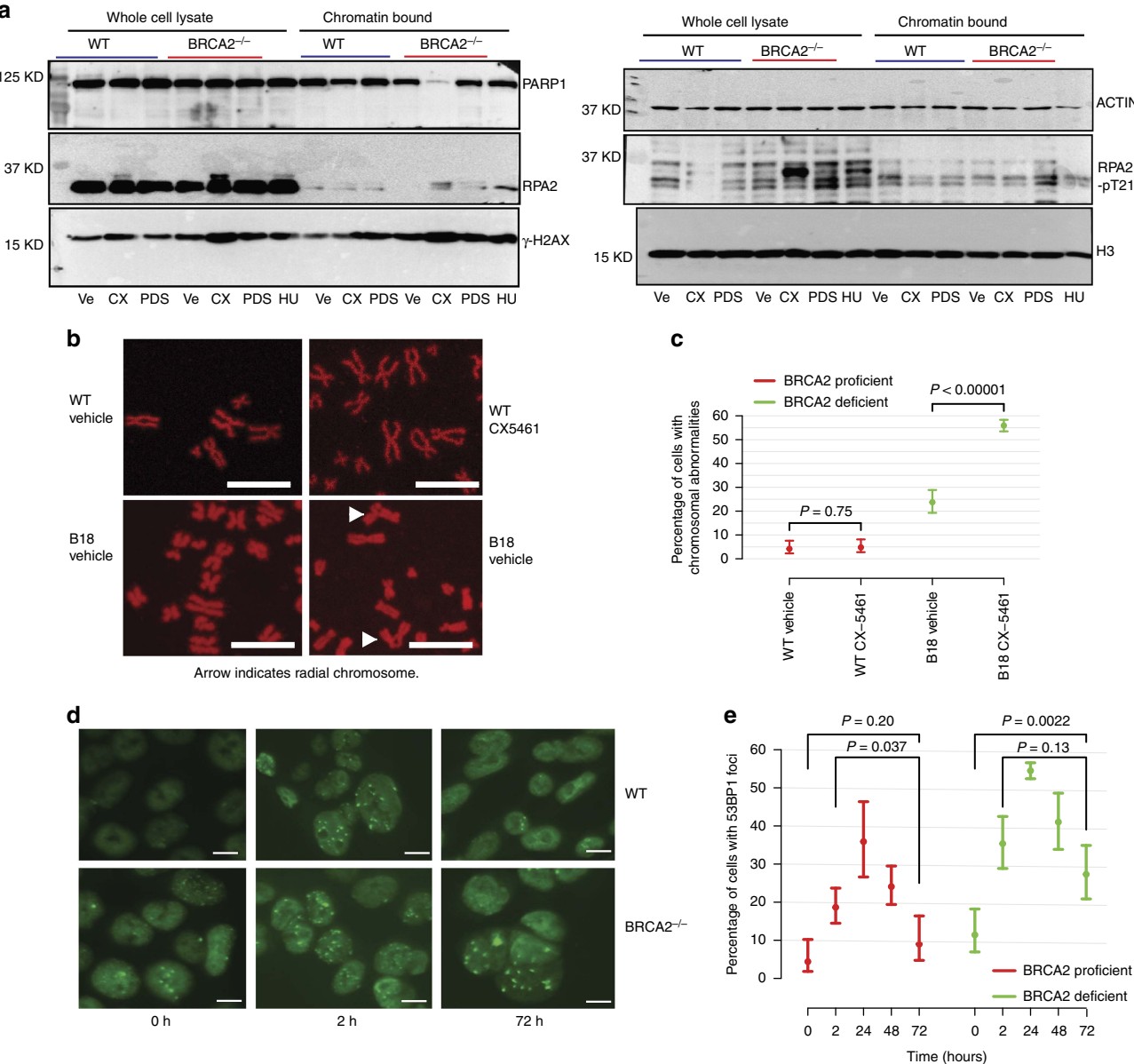

**Figure 4 | The repair of CX-5461 and CX-3543 induced DNA damage relies on BRCA pathway.** (**a**) CX-5461 induces higher levels of DNA damage in $BRCA2^{-/-}$ cells as manifested by the increase of γ-H2AX and RPA phosphorylation in $BRCA2^{-/-}$ cells. HCT116 $BRCA2^{+/+}$ and $BRCA2^{-/-}$ cells were incubated with vehicle (Ve), 10 µM CX-5461 (CX) or 10uM PDS for 4 h after 1h release from double thymidine block. Whole-cell lysates or chromatin bound fractions were analysed by Western blotting. $BRCA2^{+/+}$ cells treated with 2 mM HU for 4 h were immunoblotted as a control. Increased γ-H2AX and RPA phosphorylation happened before apoptosis as shown by the absence of Parp1 degradation. Uncropped western blotting pictures are shown in Supplementary Fig. 11. (**b**) $BRCA2^{-/-}$ HCT116 cells accumulate more chromosome abnormalities in the presence of CX-5461 ($10^{-8}$ M 48 h) demonstrated by mitotic chromosome spread. Scale bar, 10 µM. Arrows point to chromosome structure abnormalities. (**c**) Percentage of cells with chromosome abnormalities with experimental conditions stated in **b**. $N \geq 3$, >50 cells each replica. 95% CIs are shown for each data point. (**d**) 53BP foci after pulse CX-5461 treatment were resolved in WT HCT116 cells after 72 h but not in $BRCA2^{-/-}$ HCT116 cells. Cells were pulse treated with CX-5461 at $10^{-8}$ M for 2 h, and then the drug was washed out. Damage foci were monitored after 24, 48 and 72 h. Scale bar, 10 µM. (**e**) Plot displays the percentage of HCT116 cells with 53BP1 foci with experimental conditions stated in **d**. At least three independent experiments were done (>100 cells were counted each time). $P$ values were calculated using two-tailed randomization tests.

stabilizer and induces genome instability specifically at G4 sequences in both human and yeast cells.

In order to identify the targets of CX-5461 at a whole-genome level, we performed chromatin immunoprecipitation (ChIP) of RAD51 in U2OS followed by high throughput sequencing analysis (ChIP-seq), as RAD51 is able to form chromatin-bound foci in CX-5461-treated cells (Fig. 2a). We classified the G4 overlapping peaks as unique peaks (present in only one biological replicate) and reoccurring peaks (present in more than one

biological replicate). More reoccurring peaks were obtained from RAD51-ChIP under CX-5461 treatment (mean 2,816 peaks) compared with RAD51-ChIP with vehicle control (mean 65 peaks) or IgG-ChIP (mean 267 peaks) under the same concentration of CX-5461 (Fig. 6c, Supplementary Tables 8 and 9). We also found that the reoccurring peaks for RAD51-ChIP under CX-5461 treated condition contained more G4 sites (Fig. 6d, Supplementary Fig. 6d–f) per peak. These results support the notion that CX-5461 induced DNA damage is repaired by the RAD51 pathway

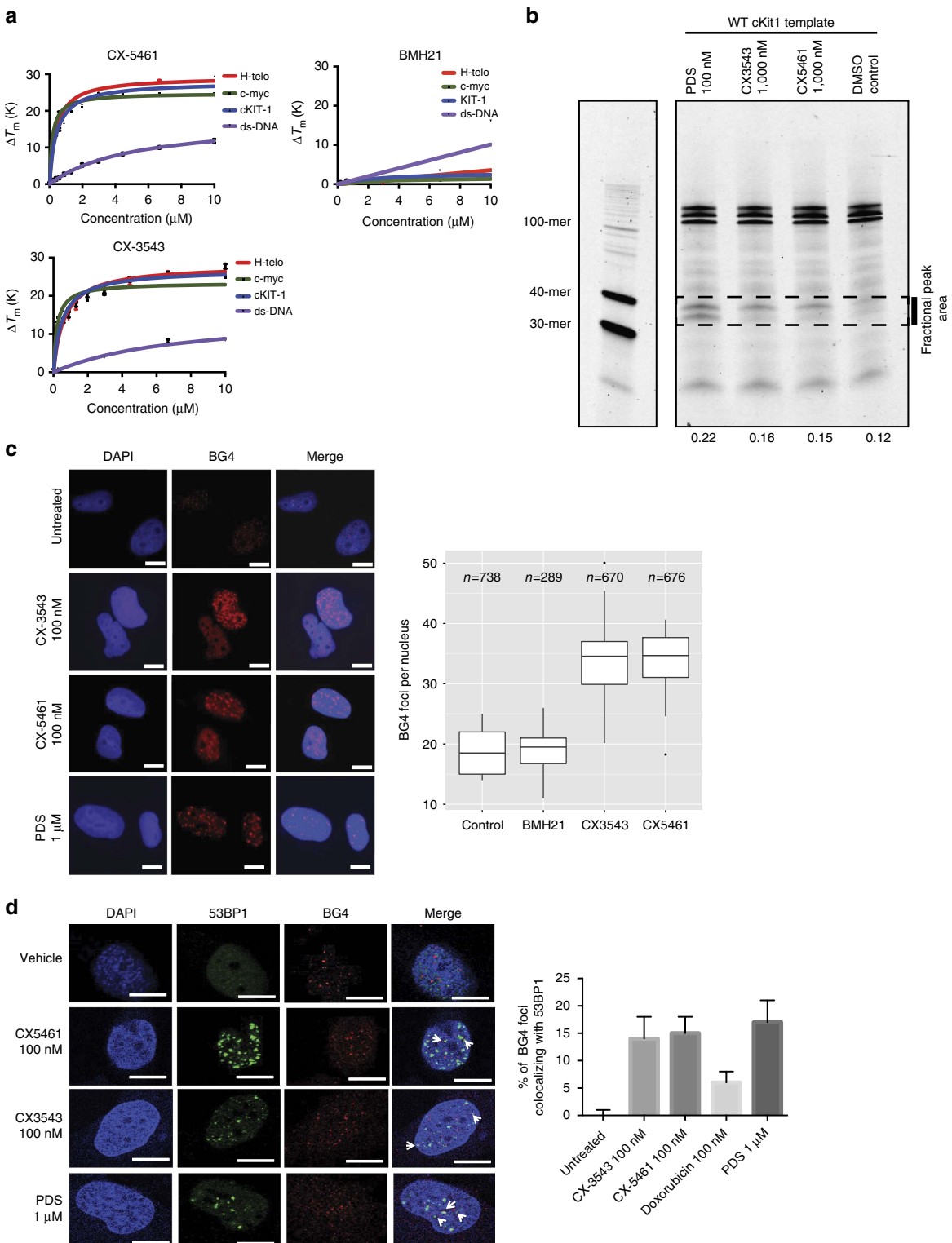

**Figure 5 | CX-5461 and CX-3543 stabilize G4 sequences.** (**a**) *In vitro* FRET melting assay with three different G4 forming DNA fragments and a non-G4 forming dsDNA control. Vertical axis, changes in melting temperature; horizontal axis, drug concentration (µM). Error bars denote the s.d.; $n = 3$. The solid lines represent the interpolation of the values with a single binding curve model. (**b**) Progression of DNA polymerase was stalled by CX-5461 and CX-3543 when incubating with G4 forming sequence cKit1. Full gel image is displayed in Supplementary Fig. 6c. (**c**) CX-5461 and CX-3543 bind to and stabilize G4 structure as demonstrated by the increased number of immunofluorescence foci with G4 binding antibody, BG4. Scale bar, 10 µM. Right panel shows the quantification. Median BG4 foci per nucleus is shown. The box extends from the 25th to 75th percentiles. (**d**) Co-localization between 53BP1 foci and BG4 foci. Drug treatment time is 24 h, $N = 2$. ~500 cells per condition were counted. Scale bar, 10 µM. Right panel shows the quantification. Error bars denote the s.d.

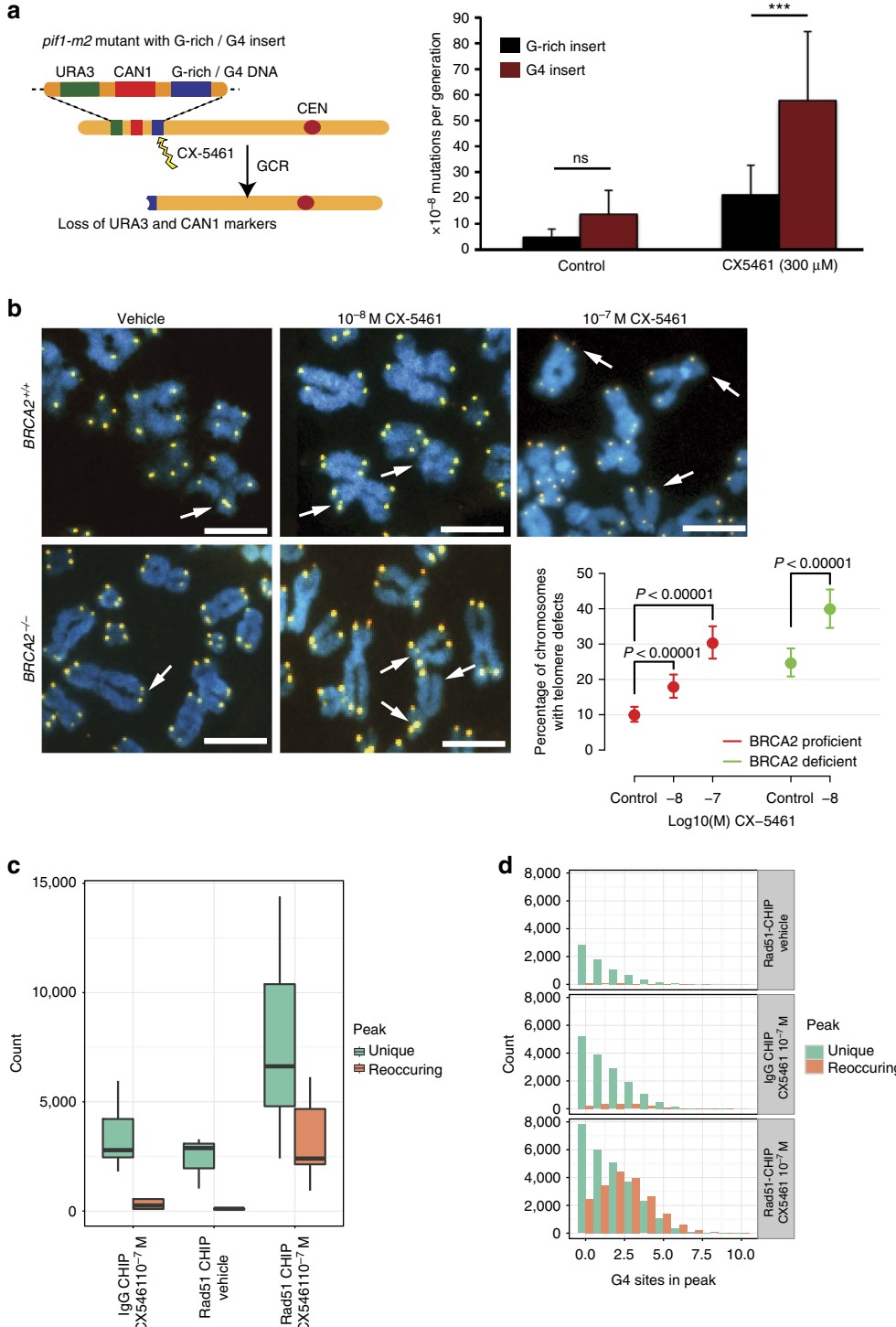

**Figure 6 | CX-5461 induces chromosome instability at G4 sequences in human and yeast systems.** (**a**) The CX-5461 induces increased GCR rates in yeast. Left panel shows the GCR assay setup. Right panel shows increased GCR rates induced by CX-5461 compared to untreated control and to a non-G4 forming G-rich control sequence (represented as per $10^{-8}$ mutations/generation). $N = 3$. (**b**) Effect of CX-5461 on telomere fragility in $BRCA2^{+/+}$ and $BRCA2^{-/-}$ HCT116 cells. Arrows point to telomere defects with either fragile telomeres or missing telomeres. $N = 3$, >100 cells each replica. The data were modelled using a logistic regression model. Scale bar, 5 µM. (**c**) RAD51-ChIP after CX-5461 treatment identified more peaks than with vehicle control and IgG-CHIP in U2OS cells. Three biological replicates per condition and two for IgG backgrounds. Peaks were classified as unique if occurred in isolation, or reoccurring if overlapped with at least one other peaks ± 500 bp. Error bars depict the range of peak numbers for three biological replicates. The exact peak numbers are shown in Supplementary Tables 8 and 9. (**d**) Peaks identified from RAD51-ChIP with CX-5461 treatment enrich for G4 sites. The number of G4 sites in unique and reoccurring peaks are shown for three ChIP conditions. G4 sites normalized by peak length are shown in Supplementary Fig. 6d. Peak length distribution is shown in Supplementary Fig. 6e. A screen shot of the peak is shown in Supplementary Fig. 6f.

and the damage loci are enriched at G4 sequences in human genome.

**Genotype specific sensitivity to CX-5461 in human and model systems**. NHEJ is another important DSB repair pathway parallel to the HR pathway. To clarify the role of the NHEJ pathway in response to CX-5461 and other G4 stabilizers, we investigated the effect of CX-5461 in cells knocked out of DNA-dependent protein kinase catalytic subunit (DNA-PKcs, encoded by *PRKDC*), which is a key component of the NHEJ pathway in mammalian cells. The IC$_{50}$ for CX-5461 decreased $\sim$seven-fold (95% CI, 2.2–22.0) in $PRKDC^{-/-}$ cells compared with $PRKDC^{+/+}$ cells (Fig. 7a, Supplementary Fig. 7a). The involvement of the NHEJ pathway in repairing G4-associated DNA damage is further strengthened by the results from LIG4 proficient and deficient isogenic cells. Both CX-5461 and PDS have higher drug sensitivity in $LIG4^{-/-}$ HCT116 cells compared with $LIG4^{+/+}$ HCT116 cells (Fig. 7b). However, consistent with the result from the paper of Zimmer *et al.*[6], knocking down 53BP1 did not affect CX-5461 sensitivity (Supplementary Fig. 7c,d). Thus, the NHEJ pathway is involved in DNA damage repair when treated with G4 stabilizers, but 53BP1 is not required for this process. Furthermore, 53BP1 and BRCA1 double knockdown cells showed reduced sensitivity to CX-5461 than did BRCA1 single knockdown cells, but as compared with non-targeting control, double knockdown cells were still more sensitive to CX-5461 (Supplementary Fig. 7d). The role of different genes in NHEJ pathway in regards to G4 resolution needs further analysis.

In order to discover additional genotypes important for CX-5461/CX-3543 drug response, we took advantage of the easy handling of drug screens in the model organism *Caenorhabditis elegans*. We examined a panel of $\sim$100 DNA damage response mutant strains in *C. elegans*, using a qualitative chronic exposure assay (Supplementary Table 2), and a quantitative acute exposure assay (Fig. 7c). In addition to *brc-2,* loss of genes involved in replication-associated repair and resolution of G4 structures (*mus-81, rfs-1, polq-1, helq-1, rtel-1*) conferred increased sensitivity to CX-5461. The Mus81 endonuclease has been implicated in the replication fork restart and the resolution of recombination intermediates[33]. The RAD51 paralog *rfs-1* and the *helq-1* helicase respond to replication fork blocking lesions but not to DNA double strand breaks[11]. *polq-1* plays a major role in microhomology-mediated end-joining and the repair of replication associated DSBs[34]. Taken together, these data support the involvement of additional DNA repair pathways in CX-drug response, as has been noted in the context of G4 structures in model organisms[35].

To further characterize the spectrum of CX-5461/CX-3543 cytotoxicity and contrast with Olaparib and BMH-21, we measured the anti-proliferative effect of CX-5461/CX-3543, BMH-21 and Olaparib across 50 well-characterised breast cancer cell lines[36] (Fig. 7d). Triple negative breast cancer (TNBC) cell lines with mutation or low expression in HR pathway genes (*BRCA1/2* and *RAD51*) (BT20, CAL51, HCC1806, HCC1395, MDA-MB-436, MDA-MB-468, HCC38)[36] (Supplementary Table 4) are more sensitive to CX-5461/CX-3543 (IC$_{50} \le 10^{-7}$ M). TNBC cell lines with deficiency (mutations or low expression) in other DNA damage repair genes, such as, *ATM, ATR* and *BARD1* also exhibit higher sensitivity to CX-5461/CX-3543 (CAL120, HCC1187, HBL100, HCC1143, HS578T, MDA-MB-231, SKBR7, SW527). Two Her2 subtype cell lines show high sensitivity to CX5461 (HCC1954, CAL-148), however both cell lines harbour a *BRCA1* mutation and low expression of FANCL in HCC1954, *RAD50*, and *PTEN* mutations in CAL-148. These results are consistent with the observation that CX-5461 is selectively active in BRCA1, BRCA2 deficient cancers, and suggest that other DNA damage repair

pathways may also result in increased sensitivity to G4 stabilizers. Furthermore, the sensitivity pattern of these breast cancer cell lines to CX-5461/CX-3543 only partially overlaps that of cisplatin and Olaparib, suggesting CX-5461/CX-3543 may be effective for chemo-resistant cancers.

**CX-5461 kills BRCA deficient and chemo-resistant PDX tumours**. The significant synthetic lethality of CX-5461 with BRCA *in vitro* motivated us to determine whether a therapeutic window could be seen *in vivo* in xenograft models. The antitumor activity of CX-5461 was evaluated in NOD/SCID/IL-2γ$^{-/-}$ immunodeficient mice with tumours subcutaneously engrafted from HCT116 isogenic cell line pairs. Strikingly, CX-5461 treatment substantially and significantly inhibited the growth rate of tumours formed from two BRCA2 deficient HCT116 cell lines (B18 and B46), in a dose-dependent manner (Fig. 8a, Supplementary Fig. 8a). Tumour growth of the BRCA2 deficient tumours remained suppressed for 10–20 days after ceasing oral dosing with CX-5461 (Fig. 8a, vertical dashed lines). In marked contrast, no tumour growth inhibition was observed on tumours derived from BRCA2 proficient HCT116 cells under the highest dosage. As a consequence, CX-5461 greatly extended the survival time of the mice bearing tumours derived from BRCA2 knockout cells (Fig. 8c). At all tested schedules, CX-5461 was well tolerated as judged by minimal effect on animal body weight (Supplementary Fig. 8a,b). Stronger tumour growth inhibition to BRCA2 deficient than BRCA2 WT cancers was also observed in xenograft model with tumours formed by DLD1 isogenic cell line pairs (Fig. 8b,d).

To address the relationship of CX-5461 activity to standard of care chemotherapy in polyclonal tumours, the activity of CX-5461 was further tested in PDX models, beginning with taxane resistant TNBC PDX tumours. Three different patient tumours were compared initially: two containing deleterious *BRCA1* or *BRCA2* mutations and one wild type status. All three patients had received a taxane prior to tumour sampling. Compared to vehicle control, CX-5461 reduced the tumour growth of all three PDX tumours, (Fig. 8e), but the inhibition of BRCA1 or BRCA2 deficient PDX tumours (CTG-0012 and CTG-0888) was much greater than on BRCA WT PDX tumour (CTG-1019). We also observed that, CTG-0012 exhibited a weak response to Olaparib but was very sensitive to CX-5461, showing that CX-5461 activity spectrum may in some cases transcend that of Olaparib. The combination effect of CX-5461 and Olaparib is similar to CX-5461 alone in these PDX models.

We next evaluated CX-5461 in a platinum-pretreated TNBC PDX (Fig. 8f). PDX CFIB-NB02 was generated from a metastatic lesion biopsy from a heavily pretreated TNBC patient (including platinum) with *BRCA1* germline missense mutation. The administration of CX-5461 resulted in dramatic tumour regression with efficacy comparable with carboplatin. PDX CFIB-70620 was generated from a TNBC patient pretreated with anthracycline/taxane chemotherapy with *BRCA2* germline mutation who had minimal response to cisplatin in the metastatic setting (Supplementary Table 3). Again, CX-5461 significantly reduced the tumour growth in this PDX model. Taken together, these data show that while CX-5461 activity spectrum partially overlaps that of PARP inhibitors and platinum salts in HR deficient tumours, CX-5461 exhibits additional activity in some tumours resistant to these agents.

**Discussion**

We have discovered two related small molecule drugs CX-5461 and CX-3543 that are able to selectively kill BRCA1/2 deficient cancer cells, one of which, CX-5461, is currently in advanced

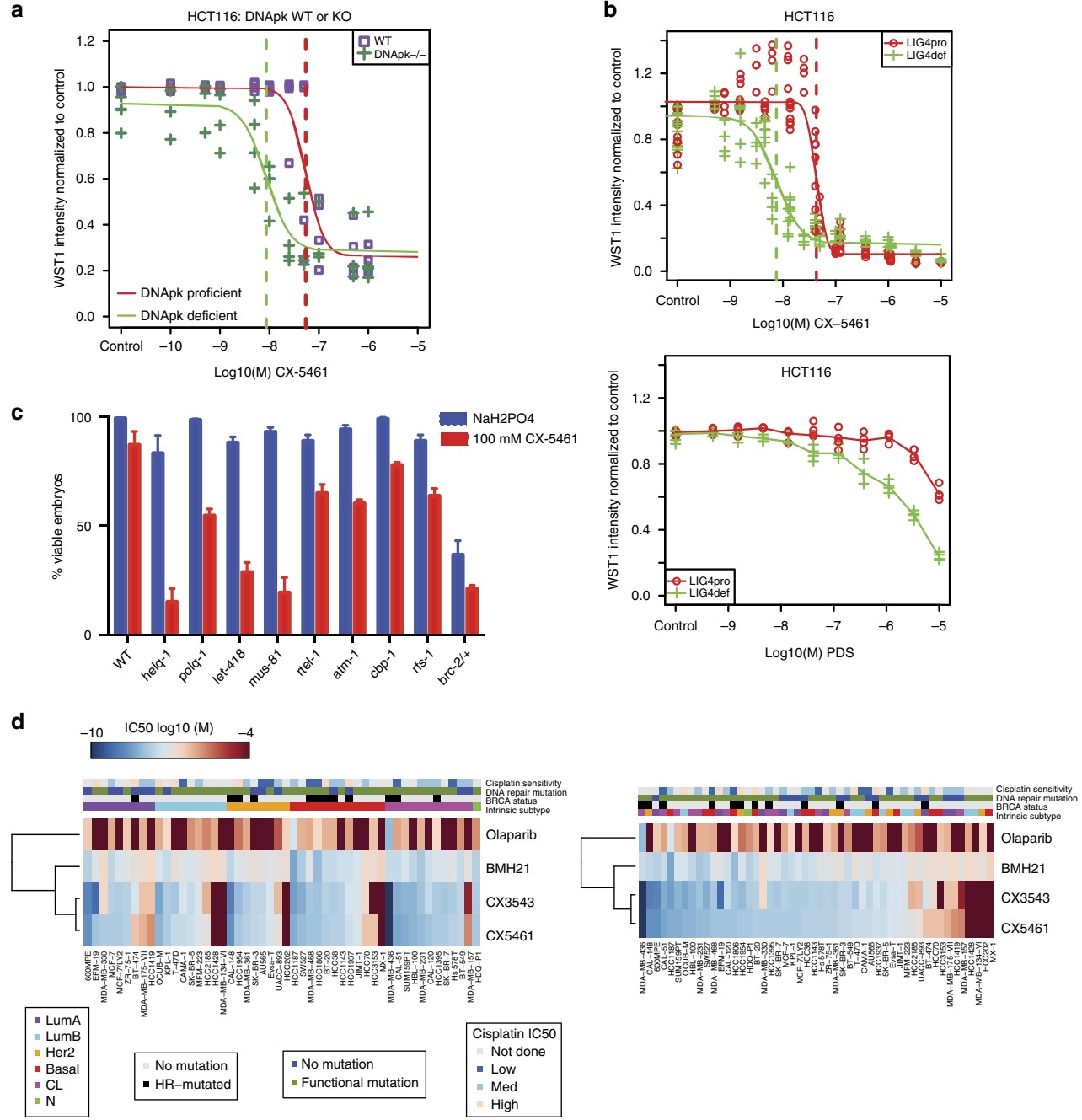

**Figure 7 | CX-5461 effectively kills tumour cells deficient for a number of DNA damage repair pathways.** (**a**) $PRKDC^{-/-}$ HCT116 cells are more sensitive to CX-5461 compared with $PRKDC$ wild type HCT116 cells. The dose sensitivity of CX-5461 (6 days in drug) was measured by WST-1 assay with representative experiment #2 shown (see Supplementary Fig. 7a for all three experiments). Vertical dashed lines indicate the estimated IC50 values. (**b**) $LIG4^{-/-}$ HCT116 cells are more sensitive to CX-5461 and PDS compared with $LIG4$ wild type HCT116 cells. The drug sensitivity (4 days in drug) was measured by a WST-1 assay. $n \geq 4$. IC50 difference for CX-5461 between $LIG4^{+/+}$ and $LIG4^{-/-}$ cells is 5.73 ($P < 10^{-10}$, $t$-test). Sigmoid fits were unavailable for PDS treated cells so no IC50 estimates or fold change was available. Assessment via a two-way ANOVA test showed that Lig4 deficient cells were more sensitive to PDS than Lig4 proficient cells ($F$-test $P < 10^{-10}$). Results of drug sensitivity to BMH-21 and cisplatin for $LIG4^{+/+}$ and $LIG4^{-/-}$ cells are shown in Supplementary Fig. 7b. (**c**) Genotype specific sensitivity to CX-5461 in $C.\ elegans$. The percentage of viable embryos observed in the progeny of animals treated for ~20 h with 100 μM CX-5461 or carrier (50 μM NaH$_2$PO$_4$). Error bars depict the s.e.m. of at least three experiments. Statistically significant difference by $t$-test was discovered for all mutants shown in this figure ($P < 0.05$) comparing carrier-treated to CX-5461-treated animals. (**d**) CX-5461/CX-3543 exhibits antiproliferative potency against a panel of breast cancer cell lines by SRB cytotoxicity assay. Heatmap represents IC50 values for 50 breast cancer cell lines treated with Olaparib, BMH-21, CX-3543 and CX-5461 for 5 days. The ward-linkage method of 'hclust' (R function) was used to compute the dendrogram. The panel on the left is ordered by intrinsic subtype of cell lines, the panel on the right is ordered by ranked value of CX-5461 sensitivity. The legends indicate the categorical values for intrinsic subtype, BRCA mutation status, DNA repair mutation status, and cisplatin sensitivity (data from Sanger cell line project).

stages of phase I safety testing in humans. This opens a new therapeutic possibility for cancers with somatic inactivation of HR pathway genes, which will be tested in a recently opened phase I/II trial in BRCA1/2 deficient patients (NCT02719977). CX-5461 is structurally related to CX-3543 (Supplementary Fig. 10) and other fluorquinolones such as QQ58, whose binding and stabilization of G4 structures by end-stacking has been determined biophysically[10]. We found that both drugs, in addition to inhibiting rRNA transcription as previously published, potently induce replication dependent DNA damage,

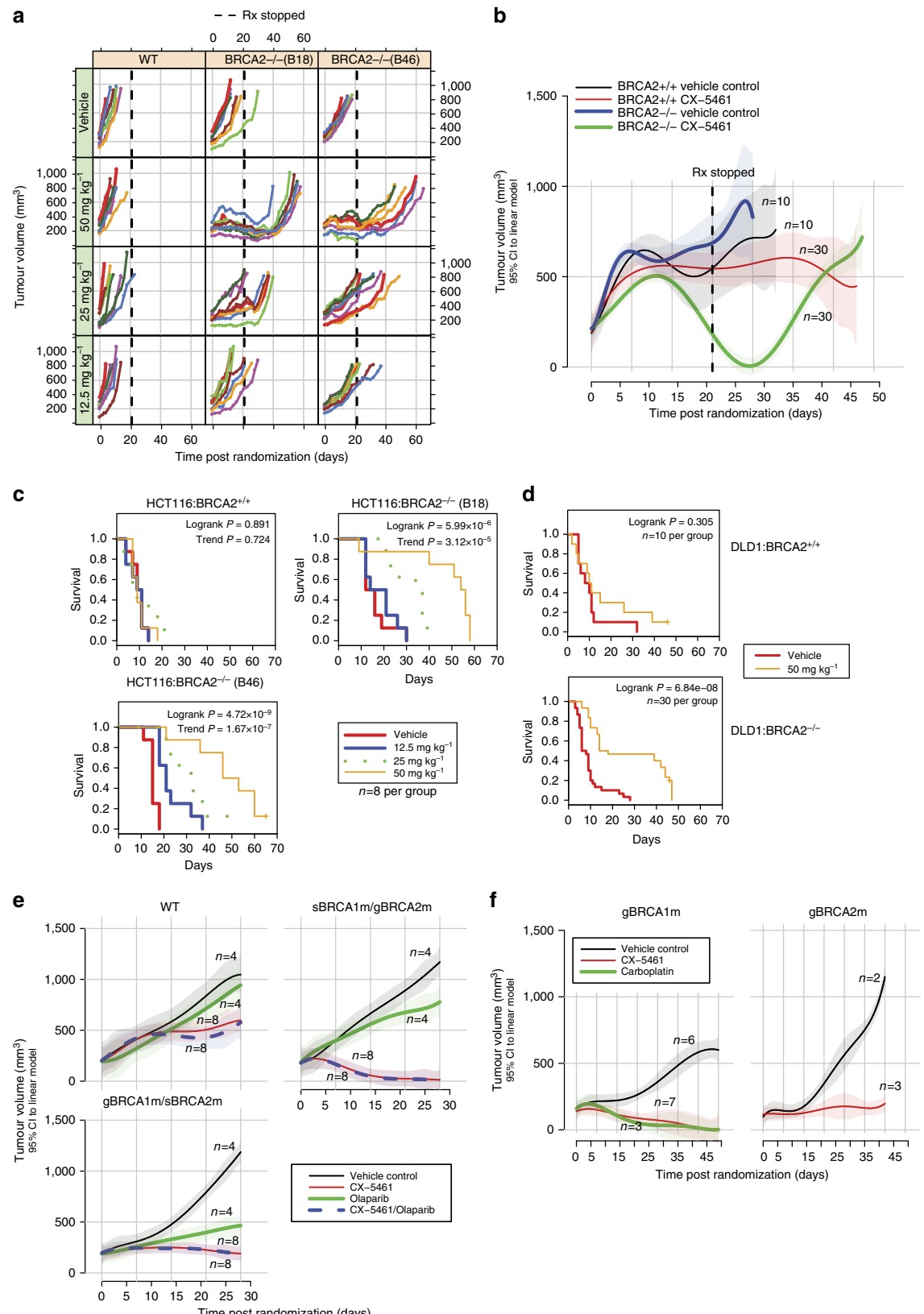

likely through binding and stabilization of G4 structure forming DNA. We note that a potent, unrelated RNA pol I inhibitor (BMH-21) which does not bind/stabilize G4 sequences, also does not induce damage or exhibit synthetic lethality, showing that RNA pol I transcription inhibition is not required for the mechanism.

Upon treatment with CX-5461 and CX-3543, G4 structures are significantly induced and accompanied by a dramatic increase of DNA damage foci in cells. BRCA deficient cells are less competent to bypass drug stabilized G4 structure during DNA replication and less efficient to repair G4 associated DNA damage. As a consequence, the accumulated DNA damage in BRCA deficient cells leads to apoptosis (Supplementary Fig. 9). Besides the HR pathway, the repair of CX-5461 and CX-3543 generated DNA damage also relies on the NHEJ pathway. We analysed CX-5461 response of three genes in the NHEJ pathway: DNA-PK, LIG4 and 53BP1. DNA-PK and LIG4 deficiency increases CX-5461 and PDS sensitivity, but 53BP1 knocking down has no effect. 53BP1 is not strictly required for NHEJ in many settings. For example, 53BP1 is required for NHEJ in class-switch recombination, but not required for NHEJ in V(D)J recombination[37,38]. It is likely that 53BP1 doesn't contribute to NHEJ of G4 associated DNA damage. In addition, some other genes in DNA replication and damage response are also involved in the repair of CX-5461/CX-3543 generated DNA damage. Mutation of ATM, ATR, BARD1, downregulation of genes in FANC pathway are associated with high efficacy to CX drugs in in vitro drug sensitivity assays. These results suggest the potential application of CX-5461 in treating cancers bearing these mutations.

The specific toxicity of CX-5461 and CX-3543 against BRCA1/2 deficient cells was seen in a number of cell lines of different genetic backgrounds (colon, breast, ovary) and different species origins (yeast, mouse and human). This is consistent with recent data using probe compounds that stabilize G4 sequences, suggesting that selective sensitivity occurs in HR deficient backgrounds[6]. CX-5461 has been engineered for superior in vivo stability and pharmacokinetics and is presently in advanced phase I trials for haematologic malignancies[15]. Consistent with the in vitro activities observed, CX-5461 exhibited a wide therapeutic index of activity in BRCA2 knockout tumour cells in xenograft models, when compared with isogenic wild type control cells. Furthermore, CX-5461 is also effective in PDX models for chemo-resistant breast cancers, including tumours relatively insensitive to PARP inhibition and/or platinum salts. Our data thus suggest immediately practical applications of CX-5461 in BRCA deficient tumours and possibly other tumours deficient for DNA repair. In particular, it is possible that the dose used to treat BRCA deficient cancers may be lower than that required to inhibit RNA polymerase I and disrupt nucleolus function, because our data suggest that BRCA deficient cells are killed by CX-5461 at low drug concentrations, which are not effective at inhibiting rDNA transcription.

In summary, our study repurposes the application of CX-5461 and CX-3543, and likely other G4 stabilizers, in treating cancers with deficiencies in BRCA pathway, NHEJ pathway, and other genes in DNA damage repair and DNA replication.

## Methods

**Human cell lines, yeast and C. elegans strains.** HCT116 $BRCA2^{+/+}$ cells and $BRCA2^{-/-}$ cells were described previously[21]. Mouse mammary tumour BRCA2 knockout cells (K14-Cre; Brca2$^{F11/F11}$; p53$^{F2-10/F2-10}$) and control mouse mammary tumour BRCA2 proficient cells (K14-Cre; Brca2$^{wt/wt}$; p53$^{F2-10/F2-10}$) were from Dr Jos Jonkers' lab and were cultured according to publication[23]. DLD1 BRCA2 proficient and BRCA2 knockout cells, HCT116 DNA-PK WT and knockout cells, LIG4 WT and knockout cells were all from Horizon Discovery and were grown in RPMI140 with 10% FBS and 2 mM L-glutamine. PEO1 and C4-2 cells were from Toshiyasu Taniguchi's lab and were grown in DMEM medium with 10% FBS and L-glutamine[22]. U2OS cells were from ATCC and were grown in McCoy's 5 A medium with 10% FBS and L-glutamine. All cell lines are mycoplasma free and have been authenticated by STR or SNP profiling.

Disease subtypes and mutation status of breast cancer cell line panel in Fig. 7d are extracted from publication[36] and Cosmic (http://cancer.sanger.ac.uk/cell_lines), and are summarized in Supplementary Table 4.

Nematode strains were maintained as described previously[39]. The strains used are listed in Supplementary Table 2. Some strains were generated by the International C. elegans Gene Knockout Consortium and the National Bioresource Project of Japan. The genotypes and background of all the yeast strains used in this study are as previously described[40].

**Cell line xenograft mouse model.** Animal procedures were approved by the University of British Columbia animal protection committee. Six to ten week old female NOD/SCID/IL-2γ$^{-/-}$ immunodeficient mice were subcutaneously engrafted with $2 \times 10^6$ tumour cells for BRCA2 proficient and $5 \times 10^6$ cells for BRCA2 knockout cells. CX-5461 was dissolved in 50 mM NaH$_2$PO$_4$, pH4 for xenograft application. Established tumours were randomized into vehicle and CX-5461-treated groups. Tumour measurement was performed by external caliper and tumour volume was calculated using the formula $[V = 1/2 \, (\text{length} \times \text{width}^2)]$. Mouse weight was measured every 3 days. CX-5461 was administered through oral gavage once every three days with three doses: 12.5 mg kg$^{-1}$, 25 mg kg$^{-1}$ and 50 mg kg$^{-1}$ for tumours formed from HCT116 cells. For mice bearing tumours from DLD1 cells, CX-5461 was administered orally at 50 mg kg$^{-1}$ once every three days. Mice were sacrificed when tumour size reached 1,000 mm$^3$, or when all other mice in a given genotype/drug dose group had been sacrificed, up to a maximum of 90 days post-xenograft.

**Patient-derived xenograft model.** BRCA status and clinical characteristic of patient tumours in Fig. 8 are listed in Supplementary Table 3. Mice PDX experiments for model CFIB-NB02 and CFIB-70620 were approved by University Health Network Research Ethics Board in Toronto #15-9481-CE. PDX experiments for Model CTG-1019, CTG-0012, and CTG-0888 were approved by Office of Laboratory Animal Welfare, NIH (A4614-01).

For Model CTG-1019 (WT), vehicle control was administered intravenously each week for 28 days (IV QWx4), Olaparib (50 mg kg$^{-1}$ dose$^{-1}$)

**Figure 8 | CX-5461 selectively suppresses growth of BRCA deficient tumours in murine xenografts and chemo-resistant PDX model. (a,b)** The effect of CX-5461 on tumor growth with xenografted tumours from isogenic WT and BRCA2 knockout HCT116 (**a**) and DLD1 (**b**) cells. For HCT116 xenograft model, three drug doses were administered at 12.5 mg kg$^{-1}$, 25 mg kg$^{-1}$ and 50 mg kg$^{-1}$ together with vehicle control. For DLD1 xenograft model, CX-5461 was administered at 50 mg kg$^{-1}$. Vertical dashed lines show the end of drug treatment. Each coloured line represents individual mouse in **a**. Solid lines in **b** represent the mean tumour volume with 95% CI (shown by shadow around solid lines) from a linear model fitted to the tumour volumes. (**c,d**) The administration of CX-5461 greatly extended the survival of mice with tumours from BRCA2 deficient but not BRCA2 proficient in xenograft models with tumours formed from HCT116 cells (**c**) and DLD1 cells (**d**). The survival of mice in experimental panel is shown as moribund-free survival time post randomization. The significance of survival differences is indicated with the log rank test (See Statistical methods). Dose-dependent trend significance is indicated with the log rank test for trend. (**e**) CX-5461 is effective for taxane resistant BRCA1/2 deficient TNBC in PDX model. Growth curve of tumours grafted from individual patients with BRCA1 and BRCA2 WT (CTG-1019), BRCA1 germline and BRCA2 somatic mutation (CTG-0888, gBRCA1m/sBRCA2m), and BRCA2 germline and BRCA1 somatic mutation (CTG-0012, sBRCA1m/gBRCA2m). Tumour volume average curves (lines) with pointwise 95% CIs (shaded regions) are shown. Mice were treated with vehicle, Olaparib, CX-5461 or the combination of Olaparib and CX-5461. See Supplementary Fig. 8c for body weight and drug dosing schedule. (**f**) CX-5461 is effective in cisplatin pretreated and BRCA1/2 deficient TNBC in PDX model. Tumour volume average curves (lines) with pointwise 95% CIs (shaded regions) are shown. Tumours are grafted from patients with BRCA1 (CFIB-NB02), and BRCA2 (CFIB-70620) germline mutation. See Supplementary Fig. 8d for body weight and drug dosing schedule.

was administered orally each day for 28 days (PO QDx28), CX-5461 (125 mg kg$^{-1}$ dose$^{-1}$) was administered intravenously each week for 14 days (IV QWx3), and Olaparib (50 mg kg$^{-1}$ dose$^{-1}$) was administered orally each day for 17 days (PO QDx17) in combination with CX-5461 (125 mg kg$^{-1}$ dose$^{-1}$) administered intravenously each week for 14 days (IV QWx3) to the groups respectively.

For Model CTG-0012 (sBRCA1m/gBRCA2m), vehicle control was administered intravenously each week for 28 days (IV QWx4), Olaparib (50 mg kg$^{-1}$ dose$^{-1}$) was administered orally each day for 28 days (PO QDx28), CX-5461 (125 mg kg$^{-1}$ dose$^{-1}$) was administered intravenously each week for 7 days (IV QWx2), and Olaparib (50 mg kg$^{-1}$ dose$^{-1}$) was administered orally each day for 10 days (PO QDx10) in combination with CX-5461 (125 mg kg$^{-1}$ dose$^{-1}$) administered intravenously each week for 7 days (IV QWx2) to the groups respectively.

For Model CTG-0888 (gBRCA1m/sBRCA2m), vehicle control was administered intravenously each week for 28 days (IV QWx4), Olaparib (50 mg kg$^{-1}$ dose$^{-1}$) was administered orally each day for 28 days (PO QDx28), CX-5461 (62.5 mg kg$^{-1}$ dose$^{-1}$) was administered intravenously each week for 28 days (IV QWx4) and Olaparib (50 mg kg$^{-1}$ dose$^{-1}$) was administered orally each day for 28 days (PO QDx28) in combination with CX-5461 (62.5 mg kg$^{-1}$ dose$^{-1}$) administered intravenously each week for 28 days (IV QWx4) to the groups respectively.

For PDX model CFIB-NB02 (gBRCA1m) and CFIB-70620 (gBRCA2m), vehicle (50 mM NaH$_2$PO$_4$, pH 4) and CX-5461 (50 mg kg$^{-1}$) were administered once every 3 days through oral gavage. Carboplatin was administered at 7.5 mg kg$^{-1}$ through IP once every week.

**siRNA sequences.** siRNA for BRCA2: Target DNA sequence: AACAA-CAATTACGAACCAAAC, dTdT overhang. Other siRNAs are ordered from Dharmacon.

**Drugs and antibodies.** Drugs and antibodies used in this study are listed in Supplementary Table 1

**WST-1 assay and clonogenic assay.** Clonogenic assay was performed as previously described[21]. Drug incubation time for clonogenic assay is 12 days. For clonogenic assay after siRNA knocking down, the cells were knocked down with siRNA one day before plating for single cells and treated with drugs. WST-1 assay was performed according to the protocol provided by manufactory (Roche, Catalogue number 11644807001). Wild type and BRCA2 knockout cells were plated in 96-well plates either untreated or treated with drug continuously during the indicated time before WST-1 assay. Drug incubation time in Fig. 1 is 6 days and in Fig. 7 is 4 days except specified. Absorbance was read with the microplate reader SpectraMax 3 (Molecular Devices, Sunnyvale, CA, USA).

**SRB assay.** Cells were seeded into 96-well plates and incubated with drugs for 5 days before fixation with 10% TCA, and being washed with tap water and air dry. Then the dried cells were stained with SRB reagent and incubated 30 min at room temperature in the dark. Plates were later washed with 1% acetic acid and completely air dry, then solubilised with 10 mM Tris. The intensity of the stained cells was read at 570 nm on spectrophotometer.

**Immunofluorescence of DNA damage foci.** For detection of RPA2, RAD51 foci cells were extracted with CSK buffer (100 mM NaCl, 300 mM sucrose, 3 mM MgCl$_2$, 10 mM PIPES (pH 6.8), with proteinase inhibitors) for 4 min before fixation. For γ-H2AX and 53BP1 foci visualization, cells were fixed with 4% paraformaldehyde in TBS (50 mM Tris-Cl, pH 7.5, 150 mM NaCl) without extraction, and then permeabilized with methanol for 1 min on ice. Cells were blocked in blocking buffer (2% BSA, 0.2% Tween-20 in TBS) for 30 min before incubation with primary antibody overnight. Secondary antibodies were incubated for 1 h. Because of background staining, damage foci positive cells were defined as cells with γ-H2AX foci≥5, 53BP1 foci≥3, RAD51 foci≥5, and RPA foci≥5.

**Immunofluorescence of G4 detection.** For detection of G4, HCT116 WT cells were treated with 100 nM of CX-5461 or CX-3543 for 24 h, then fixed with 4% paraformaldehyde in PBS (15 min) and permeabilized with 0.1% Triton-X (20 min). Cells were blocked with 5% dry milk in PBS for 1 h before incubation with primary, secondary and ternary antibody for 1 h each at 37 °C.

**Western blotting.** Cells were directly lysed in RIPA buffer with protease inhibitors and phosphatases inhibitors. Brief sonication was applied to disrupt DNA, before boiling in laemmli sample buffer for 10 min at 90 °C. 10 μg protein was loaded in each well of SDS–PAGE. For the detection of BRCA1/2 and 53BP1, 1 million cells per sample were loaded to 6% SDS–PAGE.

**Cell fractionation.** 1 million cells per sample were washed once with PBS and then lysed on ice with CSK buffer (100 mM NaCl, 300 mM sucrose, 3 mM MgCl$_2$,

10 mM PIPES (pH 6.8), 0.5% Triton X-100 with proteinase inhibitors, and phosphatase inhibitors) for 15 min. Then the cells were equally separated to three vials: one for whole cell lysate (WC), one for free cytoplasmic and nucleoplasmic proteins (FCN) and one for chromatin-bound proteins (CB). After centrifugation (900 g, 5 min at 4 °C), the FCN fraction is in the supernant and the CB fraction is in the pellet. The WC and CB fractions were resuspended in RIPA buffer with 1 × Laemmli sample buffer, then were boiled for 10–15 min before loading to SDS–PAGE.

**Apoptosis assay.** Apoptosis was assayed via Annexin V-FITC and propidium iodide (PI) staining according to the manufacturer's protocol (Invitrogen Carlsbad, CA, USA, Catalogue number, A13201). HCT116 cells were treated with CX-5461 or vehicle for 72 h; then the percentage of apoptotic cells was analysed by flow cytometry. Early apoptotic cells are Annexin V positive and PI negative (lower right quadrant in FACS profile); late apoptotic and dead cells are Annexin V and PI positive (upper right quadrant). 10,000 cells were counted for each sample.

**Cell cycle analysis.** Cells were treated with 10 μM EdU, a thymidine analogue, for 1 h prior to harvesting. Cells were then fixed with 4% paraformaldehyde for 15 min, permeabilized with 0.5% Triton-X 100 in PBS for 20 min and then incubated with click-it cocktail according to manufacturers protocol (Invitrogen Carlsbad, CA, USA, Catalogue number, C10420).

Cell cycle synchronization using a double thymidine block was carried out according to a published paper[41].

**Mitotic index.** The population of metaphase cells were identified by pH = 3 and PI double staining and analysed by flow cytometry. Cells were fixed with 70% ethanol and then blocked in TST (50 mM Tris-base, pH 7.6, 0.9% NaCl, 4% FBS, 0.2% Triton X-100) for 10 min before incubation with anti-pH 3 antibody (Millipore Cat. 06-570) for 1 h at room temperature. After washing with TST, secondary antibody was added and incubated for 1 h. Before flow cytometry, cells were incubated with 10 μg ml$^{-1}$ PI and Rnase A (100 μg ml$^{-1}$) for 10 min.

**Homologous recombination assay.** HR was measure by using DR-GFP cell line with transfected pCBA-(I-SceI)[42].

**Comet assay.** Alkaline comet and neutral comet assays were performed according to publication[26]. Images were collected by fluorescence microscope. Comets were analysed from at least 100 cells per each replica using software developed by Dr Ralph E Durand for tail moments calculation[43].

**Chromosome spread.** Cells were treated with or without CX-5461 for 2 days, and then were arrested in mitosis with 3.3 μg μl$^{-1}$ Colcemid. Cells were arrested for 2.5 h, centrifuged for 5 min at 3,500 r.p.m., resuspended in 1 ml hypotonic sodium citrate solution (0.5% Na-citrate in ddH$_2$O) at concentration of 2 × 10$^5$ cells per ml and incubated for 20 min. 500 μl – 1 ml of swelled cells were spun on positively charged slides in a Shandon 4 cytospin (900 r.p.m., high acceleration for 10 min), fixed with methanol and acetic acid mix (3:1) for 10 min and staining was performed by PI (1 μg ml) for 30 min. Spread chromosomes were visualized by fluorescence microscope at magnification of × 1,000. Chromosome abnormalities include breaks, acentric or dicentric chromosome, rings, radial chromosomes.

**Telomere FISH.** Chromosome spread was first prepared, and then telomere FISH was performed by following the protocol provided along with the Telomere PNA FISH kit from DAKO (Cat. K5326).

**DNA combing analysis.** Cells are pulse-labelled with 25 μM CldU for 30 min, then washed with pre-warmed PBS and pulsed again with 125 μM IdU for 30 min in the presence of drug. After trypsinization, cells are resuspended in PBS at 3 × 10$^6$ cells per ml. Cells are warmed at 37° for 5 min, then added an equal volume of 1% low melting point agarose solution, and poured into plug castor (BIO-RAD #170-3713). The agarose embedded with cells are solidified at 4 °C for 45 min, then are eject into ProK solution (1 mg ml$^{-1}$ proteinase K in 500 mM EDTA, 1% sarkosyl). Proteinase K pre-digested are performed at 50 °C for 30 min before use.

Each agarose plug was stained with YOYO-1 (1 μl of YOYO-1 in 150 μl TE$_{50}$) at room temperature for 30 min. Plugs were washed three times in 10 ml TE$_{50}$ and subsequently incubated in 2 ml MES (7:3 v/v MES hydrate: MES sodium salt, 50 mM, pH 5.7) and melted at 72 °C for 15 min. Silanized coverslips were incubated in the melted agarose plug solution and DNA fibres were combed onto the coverslips at 710 μm s$^{-1}$. DNA was hybridized at 60 °C for 90 min, and dehydrated sequentially in 70%, 90% and anhydrous ethanol. DNA was denatured by incubation in 1 M NaOH for 25 min at room temperature. Cover slips with combed DNA were incubating in block buffer (10% w/v albumin in PBS-T, 0.05% Tween-20, 2 mM KH$_2$PO$_4$, 2.7 mM KCl, 137 mM NaCl, pH 7.4), anti-CldU & anti-IdU solution (AbD Serotec MCA2060 and Becton Dickson 347580, respectively, 1:40 dilution in blocking buffer), anti-ssDNA solution (Millipore

MAB3034 1:50 dilution in blocking buffer), and anti-secondary solution (Alexa Fluor 546 anti-mouse IgG1, Invitrogen A21123, 1:50 dilution, Alexa fluor 648 anti-mouse IgG2, Invitrogen A21241, 1:50 dilution, and Alexa Fluor 488 anti-rat, Invitrogen A11006) for 1 h at 37 °C each time, with PBS-T washes between each staining. ProLong Gold was added to each cover slip and coverslips were stored at − 20 °C prior to imaging.

**GCR assay.** Yeast GCR assay was performed as previously described[32]. CX-5461 treatment time was 36–48 h. The GCR rates were calculated using the FALCOR web server and MMS maximum likelihood method. The number of GCR events (m) were used to determine the level of significance (P value) with a student two-tailed t-test[44].

**Yeast growth and RAD52-YFP imaging.** Yeast growth curves in YPD and Rad52-YFP imaging in synthetic complete media at 30 °C were performed as previously described[45]. The drugs were diluted in 50 mM NaH$_2$PO$_4$ buffer (CX-5461) and 1% DMSO (CX-3543).

**C. elegans CX-5461 chronic sensitivity assays.** 5 L1 larvae of each mutant strain were picked to 50 µl of M9 buffer containing OP50, carbenicillin (50 µg ml$^{-1}$), 1 × nystatin, 500 µM NaH$_2$PO$_4$ with and without 100 µM CX-5461. Worms were incubated for 4–6 days and F1 growth compared between wells with and without CX-5461. A strain was scored as sensitive if there was an obvious growth difference between wells with and without CX-5461 in three of three separate experiments.

**C. elegans CX-5461 acute sensitivity assays.** L4 stage animals were picked to fresh plates and aged for 24 h so that the animals were 1-day-old adults on the day of the experiment. Young adults were incubated in CX-5461 (in NaH$_2$PO$_4$) diluted in M9 buffer containing OP50, carbenicillin (50 µg ml$^{-1}$) and 1 × nystatin for ∼20 h. Following treatment, the animals were allowed to recover 1 h on OP50 containing NGM plates before being plated at ten per plate on NGM plates for a 4 h interval (20–24 h post-treatment). The number of embryos laid during the 4 h interval was counted and the number of arrested embryos versus hatching larvae was counted 24 h later in order to calculate the percentage of progeny surviving after treatment. All results were from at least 30 treated animals (three plates with 10 animals per plate).

**RAD51 ChIP-seq.** RAD51-ChIP was performed according to the published protocol[46] by using RAD51 antibody (Santa Cruz, Cat. 8349) after 24 h treatment with or without CX-5461 (10$^{-7}$ M). Paired FASTQ files were obtained and adaptor sequences were removed automatically with Trim Galore! (v0.4.1) [1], then aligned with bowtie2 (v2.2.7) with default settings against hg19 + rDNA. After alignment, only concordantly aligned pairs with a mapping quality of ≥30 were kept, removing misaligned and ambiguously aligned reads. Supplementary Table 8 shows the number of reads going in, successfully mapped, and remaining after filtering.

Biological replicates for treated, untreated, and IgG runs were run through MACS2 (v2.1.1) using the same replicate background untreated IgG libraries. MACS2 was set to automatically detect fragment size, build the model, and output all narrow peaks with a q-value of ≤0.25. We then categorized peaks as those that were reoccurring if it overlapped with at least one other peak (within 500 bp up and downstream) in a biological replicate with the same background library. Peaks that did not overlap with any other peaks are then referred to as 'unique' peaks. All peaks were then filtered by q-value, keeping only those with a q-value of ≤0.1.

G4 are defined as PDS-induced sites from Chambers' paper[47] on either strand that fall within the peaks ± 500 bps.

**Quantitative real-time reverse transcription PCR.** RNA isolation and reverse transcription has been described before[21]. Quantitative PCR was performed on an ABI 7900HT system. In total, 45s pre-rRNA level was measured using primers amplifying an internal transcribed spacer region as described in ref. 48 (forward primer, GCC TTC TCT AGC GAT CTG AGA G; reverse primer, CCA TAA CGG AGG CAG AGA CA) by sybrgreen analysis (Thermo Scientific, Cat. number, K0221), and also using a published primers-probe set[16] by Taqman assay (Applied Biosystems, Cat. number, 4364338). Relative cDNA amounts were normalized to ACTIN B and 18s rRNA. ACTIN B was detected by sybrgreen (forward primer, ccaaccgcgagaagatga; reverse primer, ccagaggcgtacagggatag) and Taqman RT-PCR assay (probe #64 from Roche). In total, 18s rRNA was evaluated by both sybrgreen amplification (forward primer, AAACGGCTACCACATCCA; reverse primer, CCTCCAATGGATCCTCGT, amplifying a non-spacer region in 45s pre-rRNA) and a Taqman primers-probe set (forward primer, cgattggatggttttagtgagg; reverse primer, agttcgaccgtcttctcagc; prober #81 from Roche). Actinomycin treatment (50 nm, 24 h) was used as a positive control for rDNA transcription inhibition. BRCA2 and polR1B expression were analysed by Taqman approach. BRCA2 forward primer: gcaagatggtgcagagcttta; reverse primer: cttaactgctcttcactgaaataacc; probe #84 from Roche. PolR1B forward primer: ttccagctttactgtgcaactt; reverse primer: acccagccaaccatgaca; probe #18 from Roche.

**FRET melting assay.** FRET experiments were carried out with a 200 nM oligonucleotide concentration. All DNA oligonucleotides were supplied by IBA GmbH. The donor fluorophore was 6-carboxyfluorescein (FAM) and the acceptor fluorophore was 6-carboxytetramethylrhodamine (TAMRA). The dual-labelled oligonucleotides were annealed at a concentration of 400 nM by heating at 94 °C for 10 min followed by slow cooling to room temperature at a controlled rate of 0.1 °C min$^{-1}$. 96-well plates were prepared by addition of 50 µl of the annealed DNA solution to each well, followed by 50 µl solution of molecules CX-5461, CX-3543 at the appropriate concentration.

Seven dual fluorescently labelled DNA oligonucleotides were used in these experiments: H-Telo (5′-FAM-GGG TTA GGG TTA GGG TTA GGG-TAMRA-3′), C-kit1 (5′-FAM-GGG AGG GCG CTG GGA GGA GGG-TAMRA-3′), C-myc (5′-FAM-TGA GGG TGG GTA GGG TGG GTA A-TAMRA-3′), ds-DNA (5′-FAM-TAT AGC TAT A-HEG-T ATA GCT ATA-TAMRA-3′) which is a dual-labelled 20-mer oligonucleotide comprising a self-complementary sequence with a central polyethylene glycol linker able to fold into a hairpin. The competitor was an unlabelled 26-mer oligonucleotide (5′-CAA TCG GAT CGA ATT CGA TCC GAT TG-3′). A total of 100 µM stock solutions of oligonucleotides were prepared in molecular biology grade DNase-free water. Further dilutions were carried out in 60 mM potassium cacodylate buffer, pH 7.4.

For the competition experiments, competitor was added to the fluorescently labelled quadruplex oligonucleotides sequences at an excess of 50 mol equiv. and annealed in the same solution. Measurements were made in triplicate with an excitation wavelength of 483 nm and a detection wavelength of 533 nm.

**Polymerase stop assay.** WT cKit1 template: TCT GCT TTG GGA ACC CGA GAG GAG CGC TTA TAG GGA GGG CGC TGG GAG GAG GGA GGA GAC TCA GCC GAG CAG CCG AGC ACT CTA GCT CTA G MUT cKit1 template: TCT GCT TTG GGA ACC CGA GAG GAG CGC TTA TAG AGA GAG CGC TGA GAG AGA GGA GAC TCA GCC GAG CAG CCG AGC ACT CTA GCT CTA G

Primer: [Cyanine5] CTA GAG CTA GAG TGC TCG GC

8 µl of 1 × NEB buffer 1 supplemented with 100 mM KCl containing 500 nM of template oligo and 50 nM of primer was added to a PCR tube. The solution was heated to 95 °C for 5 min then allowed to cool to room temperature. 1 µl of G4-stabilizing ligand (in a 1% DMSO/water solution) was added to the annealed material. This solution was incubated at 37 °C for 30 min. Then 1 µl of the polymerase mix (89 µl of 1 × NEB buffer 1, 10 µl of 10 mM NEB dNTP mix, and 1 µl of NEB Klenow fragment$^{exo-}$) was added. The mixture was incubated at 37 °C for 2 h. The reaction was stopped by addition of 20 µl of 2 × NEB orange gel loading dye and heated to 95 °C for 5 min. To a 15% TBE-Urea gel (Precast Novex), was added 7 µl of the quenched solution. Gel running conditions were 300 V for 35 min. The gel was scanned on a GE Typhoon imager set to detect Cy5. The lane pixel density quantitation was processed using ImageJ software. Fractional peak area was quantified by fluorescence intensity of the dashed box area over the total fluorescence on the lane. The appearance of smaller bands indicates the stall of DNA polymerase.

**Statistical methods.**

R Session information:
R version 3.2.0 (2015-04-16) was used for all statistical analysis of data.
Platform: x86_64-apple-darwin13.4.0 (64-bit)
Running under: OS X 10.10.3 (Yosemite)
locale:
[1] C
attached base packages:
[1] stats graphics grDevices utils datasets methods base
other attached packages:
[1] nlme_3.1-120 gdata_2.16.1
loaded via a namespace (and not attached):
[1] compiler_3.2.0 tools_3.2.0 grid_3.2.0 gtools_3.4.2
[5] lattice_0.20-31

**Dose response curve fitting for WST-1 assay.** Where sigmoid curves were appropriate for dose response analysis, a four-parameter logistic regression curve was fitted to the data using the R function 'nlme'. An IC50 was estimated, the difference in IC$_{50}$ values was calculated for conditions with and without drug. The set of differences was tested using the Student's paired t-test.

Where logistic model fits were not available, typically due to lack of fit when the number of drug doses assayed did not adequately cover the descending portion of the sigmoid curve, rank regression was used to compare the drug response for the multiple groups comprising the experimental condition. The entire experiment's normalized intensity values were ranked from smallest to largest, and their normalized intensity values replaced by their rank values[49]. A standard linear model of rank versus categorized drug dose was fitted for each experimental condition, with differing slope and intercept for each experimental condition, yielding a standard two-way ANOVA design. Drug dose was categorized to allow the linear model to reasonably fit the non-linear sigmoid shape of the data. Differences in response to drug dose were assessed by testing the experimental

factor using a standard *F*-test comparing the ANOVA models with and without the experimental condition variable. Dose response curves were estimated using Friedman's super-smoother[50] but $IC_{50}$s were not estimable.

**Survival analysis for xenograft study.** Mice showing tumour growth to 200 cubic millimetres were randomized to treatment groups. After randomization, mice were followed until tumour growth reached 1,000 cubic millimetres or until all other mice within each genotype had been sacrificed, up to a maximum of 90 days of observation. Thus the survival analysis endpoint was time to sacrifice due to maximum allowable tumour volume. Mice who lived without reaching maximum allowable tumour volume and were subsequently sacrificed were considered censored at their sacrifice date for the purposes of survival analysis.Survival data were displayed graphically using Kaplan–Meier curves. Time to event differences were assessed using the log-rank test[51]. Log-rank analysis and test for trend were carried out for the control group and three drug dose groups within each cell line xenograft and genotype. The trend test was conducted using the log-rank test for trend[52]. All analyses were conducted using the R statistical software package[53].

**qPCR assay statistical analysis.** qPCR Ct values were modelled using standard Gaussian linear models with categorical variables for the experimental conditions of interest, yielding two-way (for example, experiments comparing effects between BRCA2 genotypes) and three-way (for example, experiments comparing BRCA2 genotype responses within differing treatment conditions) ANOVA designs. Additional categorical variables indicating experimental plate and multiple loading controls were also included in the models, to allow model adjustment for shifts in Ct values from experiment to experiment and among loading controls. Experimental effects were assessed using *F*-tests. Effect magnitudes and 95% CIs on the $\log_2$ scale were obtained from delta-delta-Ct calculations from the linear models. Fold change estimates were obtained via exponents of the form $2^{-(\text{delta-delta-Ct})}$. When assessing multiple gene targets, resultant *F*-test *P* values were adjusted for multiple comparisons using the Benjamini–Hochberg method[54]. All analyses was conducted using the R statistical software package[53].

**Comet assay statistical analysis.** Mixed effects models compare the means of the sets of cells in each experiment and condition and obtain variance estimates reflecting between experiment and condition means rather than between individual cell measurements, with degrees of freedom reflecting the number of conditions and experiments rather than the total number of cells assayed.

Raw tail moment values were modelled using a linear mixed effects model[55]. Family-wise 95% CIs (ie, adjusted for multiple comparisons[56] for experimental condition contrasts of interest were calculated and displayed graphically.

**Data availability.** The authors declare that the dataset and computer code generated during and/or analysed during the current study are available from the corresponding author on reasonable request. RAD51-ChIP data have been deposited in the Genbank GEO database (http://www.ncbi.nlm.nih.gov/geo/) under accession codes GSE90967. G4 sequences in human genome were obtained from the paper by Chamber et al.[47], and available under the accession code GSE63874. All other remaining data are available within the Article and Supplementary Files, or available from the authors upon request.

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

## Acknowledgements

We greatly appreciate Dr Jos Jonkers' lab for generously distributing mouse mammary tumour BRCA2 knockout cells (K14-Cre; Brca2$^{F11/F11}$; p53$^{F2-10/F2-10}$) and control mouse mammary tumour BRCA2 proficient cells (K14-Cre; Brca2$^{wt/wt}$; p53$^{F2-10/F2-10}$); Toshiyasu Taniguchi's lab for sharing C4-2 and PEO1 cells; Benjamin G. Neel's lab for sharing the 50 breast cancer cell lines; Ross D. Hannan, Geraldine Aubert, and Tehmina Masud for helpful suggestions to this paper.

This work was supported by the Canadian Breast Cancer Foundation BC/Yukon, BC Cancer Foundation, Stand Up to Cancer Canada (SU2C-AACR-DT-18-15), TFRI Grant 1021, CCSRI Grant 701584, CIHR Grant MOP-126119, Canada Foundation for Innovation and Cancer Research UK. Grant Brown lab is supported by CCSRI Impact Grant 702310 (to G.W.B.) and Ontario Government Scholarship (to B.H.). S.A. is supported by a Canada Research Chair in Molecular Oncology. The Balasubramanian lab is supported by a programme grant (C14303/A17197) and core funding (C14303/A17197) from Cancer Research UK.

## Author contributions

H.X., S.A., S.B., P.H., P.C.S., D.C., T.W.M., C.C., M.B.B., J.B. G.W.B. designed the experiments. H.X., M.D.A., V.M., B.H., N.J.O., N.D.S., J.S., V.W., J.G., P.C.S., A.H.C.T., F.K., D.D. Le, D.Y., F.B.Y., J.B., A.Z., K.T., J.S., S.L., T.O., T.A., D.S., J.W. performed the experiments. S.M., D. Lai, and D.S.C. analysed the data. J.X. created *BRCA2*$^{-/-}$ HCT116 cell line. H.X., M.D.A., V.M., N.J.O. S.S. and S.A. wrote the paper.

## Additional information

**Competing financial interests:** J.S. and S.L. are employees of Senwah Biosciences Ltd. The other authors have no conflicts of interest to declare.

