## [Peer Review File · Nature Communications]

Reviewers' comments:

Reviewer #1 (Remarks to the Author):

G4 stabilizers trigger replication- and transcription-dependent DNA damage at genomic loci with G4-quadruplex forming potential. Indeed, a role for HR in resolving replication stress associated with G-quadruplexes has been demonstrated and G-quadruplexes stabilizing compounds were shown to be synthetically lethal in a BRCA1/2-deficient background (McLuckie et al J Am Chem Soc 2013; Zimmer Mol Cell 2016).

In line with these earlier studies, Xu et al report CX-5461, a molecule currently in clinical trials for hematologic malignancies, to have G4-stabilization properties and selectively targeting BRCA2-deficient cells. While the concept of G4-stabilizing compounds selectively targeting HR-deficient cells has already been demonstrated, the key point of this manuscript is the identification of a clinically relevant compound that may function through this mechanism. The authors have nicely demonstrated that the selective activity of CX-5461 in BRCA2-deficient cells is independent on the ability of these molecules to inhibit rDNA transcription. Additionally, CX-5461 was shown to display activity in BRCA-deficient tumors that had become resistant to PARPi and platinum-based regimens. The mode of action of this molecule is quite similar to what has already been reported in the literature for other G4-stabilizers. While the results are interesting and have therapeutic importance in the context of treating HR-deficient cancers, the understanding of how these molecules function in mammalian cells is not clearly presented. The results do not go beyond what is already known on how targeting G-quadruplexes can be employed as an effective strategy to target HR deficient cells or how these molecules function in the context of DNA repair. If the authors can provide a deeper mechanistic understanding of how these molecules function in mammalian cells, the findings can be potentially worthy of being accepted in Nature Communications.

Other Major points:

1. There is no direct demonstration in mammalian cells that the DNA damage observed post treatment with CX-5461/CX-3543 is due to the stabilization of G4-quadruplexes. The results (listed below) rather suggest that these G4 stabilizers (CX-5461 and CX-3543) result in DNA damage via multiple means and perhaps not only via stabilization of G4 quadruplexes:

a. Increased sensitivity of DNA PKs-/- post treatment with CX-5461/CX-3543: The role of NHEJ has not been demonstrated in resolving G-quadruplexes. In an earlier study, no effect of G4 stabilizing agents was observed in a NHEJ-deficient background (Zimmer et al,2016).

b. A marked decrease in S-phase cells after a 2-hour treatment with the inhibitors perhaps suggests that CX-5461 and CX-3543 might execute multiple effects on DNA repair and replication.

It will be important to specifically examine the replication stress at sequences reported to form G-quadruplexes. Also, does YH2Ax-staining post CX-5461/CX-3543 overlap completely with the G4 structures in the cell? Do the percentage of stabilized G4 structures that arise after CX-5461 and CX-3543 treatment increase in a BRCA-/- background?

2. Although the authors claim that that they discovered that these molecules "impede the progression of DNA replication complexes," there is no direct evidence for this in the cell-based studies. Any characterization on how these molecules are affecting replication and repair is missing. A useful approach can be using fiber-labeling experiments. These experiments would also help to reveal if these compounds are reducing the rate of replication or leading to the collapse or failed restart of the replication fork.

3. Authors need to use a more quantitative approach such as CHIP (chromatin bound RPA or RAD51) to examine if there is a difference in the levels of DDR and replication stress post treatment with CX-5461 and CX-3543 in BRCA knockout vs. WT cells.

Minor points:

1. Either use RNA pol 1 or Pol I consistently.
2. Can the CX-5461 sensitivity of the BRCA2 knockout HCT116 cell line be rescued back by an add-back of the BRCA2 copy?
3. As a control, since HCT116 is a colorectal cancer cell line, does BRCA2 knockout HCT116 exhibit a similar sensitivity to olaparib as other ovarian or breast cancer cell lines?
4. QRT-PCR is usually written as qRT-PCR
5. DNA damage response assayed by YH2AX and 53BP1 staining after CX-5461 treatment should be validated in a breast or ovarian cancer cell line.
6. Use uniform annotation for Y in YH2AX.
7. How was the mitotic index affected in BRCA^{-/-} and WT cells post treatment with CX-5461 and CX-3543?
8. Are human cells deficient in components required for resolving G-quadruplexes, such as BLM helicases, more sensitive to CX-5461?
9. Are CX-5461 and CX-3543 effective in targeting BRCA1^{-/-} 53BP1^{-/-} cells?

Reviewer #2 (Remarks to the Author):

This paper claims that two small molecule drugs in use as RNA Pol I inhibitors can be repurposed to target G-quadruplex DNA in cancer cells. The compounds act in this case by inhibition of a repair process and triggering a damage response instead of by inhibition of RNA Pol1. This novel finding of an alternate mechanism of action is important and establishes a new avenue for the development of a useful therapeutic strategy.

The paper is of particular interest and significance to those interested in G-quadruplex DNA and promises to be influential in that field. After years of languishing as a biophysical oddity, the biological significance of quadruplex DNA as a genomic regulatory element has only recently emerged and its validity as a drug target established. This paper further solidifies quadruplex DNA as a viable and specific drug target.

The paper is technically sound and generally well-written and clear. The conclusions are overall convincing and are supported by the results. The experiments (in figure 4) designed to show that CX5461 and CX3543 bind to quadruplexes are based on stabilization of the structures with respect to thermal denaturation. The shift of the T_m by more than 25 degrees is highly significant and establishes direct binding. Importantly, the authors show that the compounds have little affinity for duplex DNA, a critical consideration since in cells their quadruplex binding must compete with a vast excess of genomic DNA.

It is essential that the authors, somewhere in the paper, show and discuss the chemical structures of CX5461 and CX3543. This is important for readers to easily grasp their structural similarity and the structural features that make binding to quadruplexes favorable.

Reviewer #3 (Remarks to the Author):

The study by Xu et al., nicely demonstrate the ability of novel G4 stabilizing drugs to selectively target DNA repair defective cancers, including BRCA1, BRCA2 and DNA-PK. One of these agents is currently in clinical trials and could have immediate patient impact. G4 stabilizers are a novel class of DNA damaging agents and could be effective against many cancer types that have deficiencies in DNA repair genes.

Prior to publication Western blots should be added to Figure 1 and beyond demonstrating BRCA +/- status of cell lines. In cases where siRNA is used Western blots should be shown for protein knockdown and to confirm the effects are not off-target.

In figure 1G why is rDNA transcription inhibited better in BRCA proficient cells?

The Figures are not in the order that they are mentioned in the text. Please re-order the Figures - both main Figures, Supplementary and Tables so that they appear in the order they are mentioned in the text. The current format it is very difficult to follow.

We thank the reviewers for their comments which we have addressed in a revised manuscript with new data included, after several months of work. Our detailed responses are noted below.

The major elements new data added to the paper include:

- (1) Quantification of replication fork stalling with DNA combing, to show that CX-5461 induces fork stalls in wt and BRCA2 deficient cells
 - (2) Comparison of RPA expression and phosphorylation after CX-5461 treatment in WT and *BRCA2*^{-/-} cells.
 - (3) Colocalization of BG4 and 53BP1 foci after CX-5461 treatment and comparing with random DNA damaging agents
 - (4) Telomere FISH results post CX-5461 exposure support CX-5461 induces DNA damage specifically at G4 forming sequences in human genome.
 - (5) A Chip-seq experiment with RAD51 to show that CX-5461 exposure leads to DNA damage enriched at G4 containing sequencing in human cells
 - (6) Quantification of the sensitivity of *LIG4*^{-/-} cells to CX-5461, and the result further supports the involvement of NHEJ in the repair of G4 associated DNA damage.
-

Reviewers' comments:

Reviewer #1 (Remarks to the Author):

G4 stabilizers trigger replication- and transcription-dependent DNA damage at genomic loci with G4-quadruplex forming potential. Indeed, a role for HR in resolving replication stress associated with G-quadruplexes has been demonstrated and G-quadruplexes stabilizing compounds were shown to be synthetically lethal in a BRCA1/2-deficient background (McLuckie et al J Am Chem Soc 2013; Zimmer Mol Cell 2016).

In line with these earlier studies, Xu et al report CX-5461, a molecule currently in clinical trials for hematologic malignancies, to have G4-stabilization properties and selectively targeting BRCA2-deficient cells. While the concept of G4-stabilizing compounds selectively targeting HR-deficient cells has already been demonstrated, the key point of this manuscript is the identification of a clinically relevant compound that may function through this mechanism. The authors have nicely demonstrated that the selective activity of CX-5461 in BRCA2-deficient cells is independent on the ability of these molecules to inhibit rDNA transcription. Additionally, CX-5461 was shown to display activity in

BRCA-deficient tumors that had become resistant to PARPi and platinum-based regimens. The mode of action of this molecule is quite similar to what has already been reported in the literature for other G4-stabilizers. While the results are interesting and have therapeutic importance in the context of treating HR-deficient cancers, the understanding of how these molecules function in mammalian cells is not clearly presented. The results do not go beyond what is already known on how targeting G-quadruplexes can be employed as an effective strategy to target HR deficient cells or how these molecules function in the context of DNA repair. If the authors can provide a deeper mechanistic understanding of how these molecules function in mammalian cells, the findings can be potentially worthy of being accepted in Nature Communications.

Rebuttal: We think our manuscript goes well beyond published papers for the following reasons:

- 1) This is the first time a G4 stabilizing small molecule with structure, PK and phase 1 safety data – i.e. capability of actually being trialed in humans - been shown to reduce the growth of polyclonal patient derived tumors. This is the main point of the manuscript – the translational impact.
- 2) We now show that replication fork stalling by CX-5461 can indeed be observed in combing assays. (new data)
- 3) We include new experimental data that shows G4 stabilizer induced DNA damage at a whole-genome level by RAD51 ChIP-seq and at a single cell imaging level with 53BP1 foci and BG4 foci colocalization. Results from these two experiments show that CX-5461 induced DNA damage happens at loci enriched with G4 sequences. This substantially supports the mechanism of action for this clinical compound.
- 4) We show that besides the BRCA pathway, the NHEJ pathway also contributes to the repair of G4 associated DNA damage. In contrast with previous papers which have mentioned only 53BP1, we explored additional genes in the NHEJ pathway (DNA-PK, 53BP1 and Lig4) than in Zimmer's paper¹. In addition, through a *C.elegans* screen, we found that other genes in DNA replication and repair pathways are also potential targets of G4 stabilizers. (new data)
- 5) We define that the RNA pol I activity of CX-5461/CX-3543 is not required for their synthetic lethality with BRCA1/2.

Other Major points:

1. There is no direct demonstration in mammalian cells that the DNA damage observed post treatment with CX-5461/CX-3543 is due to the stabilization of G4-quadruplexes. The results (listed below) rather suggest that these G4 stabilizers (CX-5461 and CX-3543) result in DNA damage via multiple means and perhaps not only via stabilization of G4 quadruplexes:

a. Increased sensitivity of DNA PKs^{-/-} post treatment with CX-5461/CX-3543:

The role of NHEJ has not been demonstrated in resolving G-quadruplexes.

In an earlier study, no effect of G4 stabilizing agents was observed in a NHEJ-deficient background (Zimmer et al,2016).

Rebuttal: (new data in Figure 5B) The increased sensitivity of DNA-PK deficient cells to a well recognized G4 stabilizer, pyridostatin has been published before². Zimmer et al reported that 53BP1 deficiency did not increase sensitivity to PDS¹, and concluded that NHEJ pathway is not required for G4 associated DNA damage repair. However, 53BP1 is not strictly required for NHEJ in many physiological settings. For example, 53BP1 is required for NHEJ in CSR, but not required for NHEJ in the context of V(D) J recombination^{3,4}. It is likely that 53BP1 does not contribute to NHEJ of G4 associated DNA damage. We report that both *DNA-PK^{-/-}* and *LIG4^{-/-}* cells exhibit increased drug sensitivity to CX-5461 and PDS, supporting the involvement of NHEJ pathway in repairing G4 induced DNA damage. The drug sensitivity results of *LIG4^{-/-}* cells to CX-5461 and PDS have been added to the revised manuscript in Figure 5B.

b. A marked decrease in S-phase cells after a 2-hour treatment with the inhibitors perhaps suggests that CX-5461 and CX-3543 might execute multiple effects on DNA repair and replication. It will be important to specifically examine the replication stress at sequences reported to form G-quadruplexes.

Rebuttal: There are now several experiments in the manuscript supporting the presence of replication stress at G4 forming sequences.

(i) (new data in Figure 4F) In human cells (HCT116) we assayed the integrity of telomeres, which are loci enriched with G4 sequences. Our Telo-FISH results show that in the presence of CX-5461 telomere defects occurred at a significantly higher level than vehicle control, supporting the notion that CX-5461 increases genome instability at endogenous G4 forming loci in the human genome.

(ii) The results from a polymerase stop assay demonstrate that CX-5461/CX-

3543 increase DNA polymerase stalling specifically at a G4 site, an endogenous sequence from cKit promoter (Figure 4B).

(iii) By using a GCR assay in yeast, we compared the contribution of a G4 forming sequence to GCR versus a non-G4 forming control sequence. The results clearly show that the G4 sequence induced higher level of GCR than the G-rich but not G4 forming sequence (Figure 4D).

Also, does γ H2Ax-staining post CX-5461/CX-3543 overlap completely with the G4 structures in the cell? Do the percentage of stabilized G4 structures that arise after CX-5461 and CX-3543 treatment increase in a BRCA-/- background?

Rebuttal:

(i) (new data in Figure 4E) γ -H2AX staining will not be a stringent co-localization experiment because of its broad spreading on chromosomes that can occur megabases away from the site of double strand breaks (and relevant G4 targets). Instead, we performed co-localization for 53BP1 foci and BG4 foci, shown in Fig 4E. 53BP1 has a narrower domain of spread on recruitment to damage sites. We found significantly increased co-localization of damage foci (53BP1) and BG4 G4 foci in CX-5461/CX-3543 treated cells at a level similar to PDS, in contrast to “random” genome damage induced by a non-G4 related compound, doxorubicin. This demonstrates a significant increase in colocalised structures with CX-5461 treatment, which supports the notion that damage is occurring at some set of G4 sites in mammalian cells. In *BRCA2*^{-/-} cells, G4 structures identified through BG4 IF occur at a level similar to that seen in WT cells (Figure R3), suggesting the genotype per se is not generating more G4s to start with.

The overlap between BG4 foci (Figure 4E) and 53BP1 foci is not complete (15%), but it is not clear why it would be expected to be so. G4 structures form transiently even with stabilizers. G4 sequences might have been deleted before DNA damage proteins bind to the damaged DNA. On the other hand, not all stabilized G4 structures will necessarily lead to damage. Additional factors are that: (i) immunofluorescence with BG4 antibody, the method for detecting stable G4 *in vivo*, can't capture all known G4 sites. BG4 IF sensitivity will not be sufficient to detect single quadruplexes, but rather only generate sufficient signal for fluorescence detection at clusters of quadruplexes⁵. (ii) there are pathways independent of 53BP1 to repair G4 associated DNA damage. (iii) the distribution of distance between damage and G4 sites is not

understood and some proportion of the damage could be downstream of the G4. All these factors could preclude high overlap co-localization of DNA damage loci and BG4 loci. What is important is that the degree of overlap shows dynamics to CX5461 and is much greater than when random damage is induced. The Balasubramanian lab has performed co-localization on other common G4 stabilizers and found similar level of co-localization as CX drugs, suggesting that this is a common feature for currently available G4 stabilizers.

(ii) (new data in Figure 4G, 4H) We have included the results of a RAD51 ChIP-seq experiment, (conducted with 3 biological replicates per condition) which shows enrichment of G4 containing sequences upon drug exposure, when compared with non-drug treated RAD51 ChIP-seq result under the same conditions. This is consistent with the notion that damage is occurring at some subset of G4 sites.

The reviewer mentions that damage at sites other than G4 may be possible. We do not exclude this in our manuscript, rather we show that stabilization of G4 structures is one likely and prevalent mechanism of damage. This discussion point has been added to the text.

2. Although the authors claim that that they discovered that these molecules "impede the progression of DNA replication complexes," there is no direct evidence for this in the cell-based studies. Any characterization on how these molecules are affecting replication and repair is missing. A useful approach can be using fiber-labeling experiments. These experiments would also help to reveal if these compounds are reducing the rate of replication or leading to the collapse or failed restart of the replication fork.

Rebuttal: (new data in Figure 3D) We performed a DNA combing assay and have added the results to the manuscript. The DNA combing assay result revealed a reduced replication rate in the presence of CX-5461, and further supported the importance of BRCA2 in bypassing CX-5461 induced obstacles to DNA replication.

3. Authors need to use a more quantitative approach such as CHIP (chromatin bound RPA or RAD51) to examine if there is a difference in the levels of DDR and replication stress post treatment with CX-5461 and CX-3543 in BRCA knockout vs. WT cells.

Rebuttal: We have added additional data to address this comment. In response

we note

- (i) Rad51 is not able to bind to chromatin without BRCA2. Therefore, using chromatin bound RAD51 to compare the levels of DDR between *BRCA* knockout and WT cells is not applicable.
- (ii) (new data) We compared the level of RPA in whole cell lysate and chromatin bound fraction in *BRCA2* knockout vs. WT cells post treatment with CX5461 by Western blotting. Increased γ -H2AX and RPA phosphorylation can be found in *BRCA2* knockout HCT116 cells in both whole cell lysate and chromatin bound fractions, although the phosphorylation of RPA at chromatin bound fraction is harder to be captured by Western blotting. These results have been added to the manuscript (Figure 3E).
- (iii) (new data) The results of chromosome abnormalities identified from chromosome spreads (Figure 3F) and Telo-FISH (Figure 4F) also show quantitative comparison on the level of DDR between WT and *BRCA2* knockout cells post treatment with CX-5461/CX-3543.
- (iv) (new data) The DNA combing assay directly compared replication stress between *BRCA2* knockout vs. WT HCT116 cells after CX-5461 treatment. Figure 3D shows that DNA replication rate in *BRCA2*^{-/-} cells were immediately reduced (30 mins post CX-5461 treatment) and at a higher level than WT cells.
- (v) In Figure 2C of our original paper, DDR levels were quantified and compared between *BRCA2* knockout and WT cells by using 53BP1 foci as the readout. Figure 2C clearly shows that *BRCA2* knockout cells accumulate more 53BP1 foci compared with WT cells, especially at low CX5461 and CX3543 concentrations (10^{-7} M and 10^{-8} M).
- (vi) FACS results demonstrate that active replication (EdU staining population) decreases more in *BRCA2* knockout cells than in WT cells (Figure 3A, Figure S5A), suggesting more replication stress in *BRCA2* knockout cells.

Minor points:

1. Either use RNA pol 1 or Pol I consistently.

Rebuttal: Thank you for pointing this out and we have made changes in the manuscript accordingly.

2. Can the CX-5461 sensitivity of the BRCA2 knockout HCT116 cell line be rescued back by an add-back of the BRCA2 copy?

Rebuttal: This is already shown in the paper. Ovarian cancer cell line PEO1 is a BRCA2 loss of function cell line with a hemizygous nonsense mutation in *BRCA2*. C4-2 is originated from PEO1, but has a secondary single base pair substitution in *BRCA2* that changes the stop codon into amino acid coding triplet restoring the expression of BRCA2. We have shown in Figure 1E that C4-2 cells have reduced sensitivity to CX-5461 compared with PEO1 cells. This result supports the notion that CX-5461 sensitivity in BRCA2 deficient cells can be rescued by the re-expression of BRCA2.

3. As a control, since HCT116 is a colorectal cancer cell line, does BRCA2 knockout HCT116 exhibit a similar sensitivity to olaparib as other ovarian or breast cancer cell lines?

Rebuttal: (new data in Figure R1) Higher sensitivity to Olaparib treatment is also observed in *BRCA2* knockout HCT116 cells. Clonogenic analysis of Olaparib sensitivity in WT and *BRCA2*^{-/-} HCT116 cells has been published by us in a previous paper⁶. We also performed a WST-1 assay and analyzed the difference of Olaparib sensitivity between WT and *BRCA2*^{-/-} HCT116 cells statistically. The WST-1 result is shown in Figure R1.

4. QRT-PCR is usually written as qRT-PCR

Rebuttal: Thank you for pointing this out and we have made changes in the manuscript accordingly.

5. DNA damage response assayed by YH2AX and 53BP1 staining after CX-5461 treatment should be validated in a breast or ovarian cancer cell line.

Rebuttal: (new data in Figure R2) We validated DNA damage foci formation in a breast cancer cell line CAL-120 and have included the result in supplementary Figure R2.

6. Use uniform annotation for Y in YH2AX.

Rebuttal: Thank you for pointing this out and we have made changes in the manuscript accordingly

7. How was the mitotic index affected in BRCA^{-/-} and WT cells post treatment with CX-5461 and CX-3543?

Rebuttal: (new data in Figure S5B) The mitotic index was greatly decreased with CX-5461 treatment in both *BRCA2*^{-/-} and WT cells.

8. Are human cells deficient in components required for resolving G-quadruplexes, such as BLM helicases, more sensitive to CX-5461?

Rebuttal: (new data in Figure R4) We compared CX-5461 sensitivity in *BLM*^{+/+} and *BLM*^{-/-} cells, but failed to discover a significant difference. Although BLM has the ability to resolve G4 structures, it may not be able to resolve CX-5461 stabilized G4 structures. In addition, helicases are extremely redundant in human cells. Knocking out only one helicases might not reveal any significant effect.

9. Are CX-5461 and CX-3543 effective in targeting BRCA1^{-/-} 53BP1^{-/-} cells?

Rebuttal: (new data in Figure S7D) We did BRCA1 and 53BP1 single and double knock down in HCT116 cells. Drug sensitivity results revealed that 53BP1 deficiency decreases CX-5461 sensitivity in BRCA1 knockdown cells. However, compared with non-targeting control, double knockdown cells are still more sensitive to CX5461. This result has been included in supplementary Figure S7D.

Reviewer #2 (Remarks to the Author):

This paper claims that two small molecule drugs in use as RNA Pol I inhibitors can be repurposed to target G-quadruplex DNA in cancer cells. The compounds act in this case by inhibition of a repair process and triggering a damage response instead of by inhibition of RNA Pol1. This novel finding of an alternate mechanism of action is important and establishes a new avenue for the development of a useful therapeutic strategy.

The paper is of particular interest and significance to those interested in G-quadruplex DNA and promises to be influential in that field. After years of

languishing as a biophysical oddity, the biological significance of quadruplex DNA as a genomic regulatory element has only recently emerged and its validity as a drug target established. This paper further solidifies quadruplex DNA as a viable and specific drug target.

The paper is technically sound and generally well-written and clear. The conclusions are overall convincing and are supported by the results. The experiments (in figure 4) designed to show that CX5461 and CX3543 bind to quadruplexes are based on stabilization of the structures with respect to thermal denaturation. The shift of the T_m by more than 25 degrees is highly significant and establishes direct binding. Importantly, the authors show that the compounds have little affinity for duplex DNA, a critical consideration since in cells their quadruplex binding must compete with a vast excess of genomic DNA.

It is essential that the authors, somewhere in the paper, show and discuss the chemical structures of CX5461 and CX3543. This is important for readers to easily grasp their structural similarity and the structural features that make binding to quadruplexes favorable.

Response: We have added the structures and some discussion of them to the manuscript.

Reviewer #3 (Remarks to the Author):

The study by Xu et al., nicely demonstrate the ability of novel G4 stabilizing drugs to selectively target DNA repair defective cancers, including BRCA1, BRCA2 and DNA-PK. One of these agents is currently in clinical trials and could have immediate patient impact. G4 stabilizers are a novel class of DNA damaging agents and could be effective against many cancer types that have deficiencies in DNA repair genes.

Prior to publication Western blots should be added to Figure 1 and beyond demonstrating BRCA +/- status of cell lines. In cases where siRNA is used Western blots should be shown for protein knockdown and to confirm the effects are not off-target.

Rebuttal: (new data in Figure S7C) The expression of BRCA2 in *BRCA2*^{+/-} and *BRCA2*^{-/-} HCT116 cells have been shown in our previously published paper ⁶. Western blotting showing protein knocking down for BRCA1, BRCA2 and 53BP1 is demonstrated in Figure S7C. For each specific gene knocking down, we have experimental results showing the corresponding phenotypes

consistent with the function of that specific gene. After BRCA1 or BRCA2 knockdown, HR rate was reduced as shown in Figure S2F. For 53BP1 knockdown, we showed the reduced sensitivity to Olaparib in 53BP1 and BRCA1 double knockdown cells comparing with BRCA1 single knockdown. For polR1B knockdown, we showed the knockdown cells have lower level of 45S pre-rRNA (Figure S3C). We didn't find a good antibody for PolR1B, instead, we showed the decrease of PolR1B by RT-PCR (Figure S3C).

In figure 1G why is rDNA transcription inhibited better in BRCA proficient cells?

Rebuttal: For most of the conditions in Figure 1G, rDNA transcription inhibition is similar between BRCA2 proficient and deficient cells. Only when cells are treated with 10^{-5} M CX5461 for 24 hours, and when cells were treated with actinomycin for 24 hours, did BRCA2 proficient cells show higher rDNA transcription inhibition as shown in Figure 1G. We also assayed the effect of rDNA transcription inhibition 2 hours after drug treatment, and no statistical difference was discovered between BRCA2 proficient and deficient cells (Figure S3H).

The mechanism of the difference in rDNA transcription inhibition is not known. However, only under long time duration (24 hours) and higher drug concentration (10^{-5} M), did BRCA2 deficient cells show reduced rDNA transcription inhibition. A possible mechanism might be that the reduced inhibition of rDNA transcription in BRCA2 deficient cells is a stress response.

The Figures are not in the order that they are mentioned in the text. Please re-order the Figures - both main Figures, Supplementary and Tables so that they appear in the order they are mentioned in the text. The current format it is very difficult to follow.

Rebuttal: Thank you for pointing this out and we have made changes in the manuscript accordingly.

Figure R1: *BRCA2* knockout HCT116 cells are more sensitive to Olaparib than WT HCT116 cells by WST-1 assay. Cells were incubated with Olaparib for 96 hrs. . All 3 WST-1 assay results are shown. The mean IC50 difference between WT and *BRCA2* knockout cells towards Olaparib across the 3 experiments was 155.41.

Figure R2: CX-5461 induces DNA damage foci (53BP1 foci) in a breast cancer cell line CAL-120. Cells were treated with vehicle or different concentrations of CX-5461 for 2 hours before processed for IF. The data were modeled using a logistic regression model, from which percentage estimates, 95% confidence intervals, and likelihood ratio p-values were obtained.

Figure R3: WT and *BRCA2*^{-/-} cells show similar levels of BG4 staining. Numbers of BG4 foci within each cell were shown. Error bars denote standard deviation.

Figure R4: Clonogenic assay results for HCT116 BLM proficient and deficient cells treated with CX-5461 or Cisplatin for 10 days. An omnibus test of genotype effect yields an F-test statistic with p=0.14. We fail to reject the null hypothesis of no genotype difference in drug response. These data do not provide evidence to suggest that human cells deficient in BLM helicase components are more sensitive to CX-5461.

1. Zimmer, J. *et al.* Targeting BRCA1 and BRCA2 Deficiencies with G-Quadruplex-Interacting Compounds. *Molecular Cell* **61**, 449–460 (2016).
2. McLuckie, K. I. *et al.* G-quadruplex DNA as a molecular target for induced synthetic lethality in cancer cells. *Journal of the American Chemical Society* **135**, 9640–9643 (2013).
3. Manis, J. P. *et al.* 53BP1 links DNA damage-response pathways to immunoglobulin heavy chain class-switch recombination. *Nature Immunology* **5**, 481–487 (2004).
4. Ward, I. M. *et al.* 53BP1 is required for class switch recombination. *J. Cell Biol.* **165**, 459–464 (2004).
5. Biffi, G., Tannahill, D., McCafferty, J. & Balasubramanian, S. Quantitative visualization of DNA G-quadruplex structures in human cells. *Nature Chemistry* **5**, 182–186 (2013).
6. Xu, H. *et al.* Up-regulation of the interferon-related genes in BRCA2 knockout epithelial cells. *The Journal of pathology* **234**, 386–397 (2014).

REVIEWERS' COMMENTS:

Reviewer #1 (Remarks to the Author):

The authors have substantially strengthened the paper and distinguished it from prior reports. The work is nicely executed and represents a greater conceptual advance. It should be well received and I am in favor of acceptance.

Reviewer #3 (Remarks to the Author):

The issues have been addressed and this work is acceptable for publication.